# Expected flow networks in stochastic environments and two-player zero-sum games

**Marco Jiralerspong**[*], **Bilun Sun**[*], **Danilo Vucetic**[*], **Tianyu Zhang**,
**Yoshua Bengio**[◇], **Gauthier Gidel**[†], **Nikolay Malkin**
Mila – Québec AI Institute, Université de Montréal
$\{$marco.jiralerspong,bilun.sun,danilo.vucetic,tianyu.zhang, yoshua.bengio,gidelgau,nikolay.malkin$\}$@mila.quebec

## Abstract

Generative flow networks (GFlowNets) are sequential sampling models trained to match a given distribution. GFlowNets have been successfully applied to various structured object generation tasks, sampling a diverse set of high-reward objects quickly. We propose expected flow networks (EFlowNets), which extend GFlowNets to stochastic environments. We show that EFlowNets outperform other GFlowNet formulations in stochastic tasks such as protein design. We then extend the concept of EFlowNets to adversarial environments, proposing adversarial flow networks (AFlowNets) for two-player zero-sum games. We show that AFlowNets learn to find above 80% of optimal moves in Connect-4 via self-play and outperform AlphaZero in tournaments.
Code: `https://github.com/GFNOrg/AdversarialFlowNetworks`.

## 1 Introduction

Generative flow networks (GFlowNets; Bengio et al., 2021; 2023; Lahlou et al., 2023) are a unifying algorithmic framework for training stochastic policies in Markov decision processes (MDPs; Sutton & Barto, 2018) to sample from a given distribution over terminal states. GFlowNets have been used as an efficient alternative to Monte Carlo methods for amortized sampling in applications that require finding diverse high-reward samples (Jain et al., 2022; Zhang et al., 2022; 2023a; Li et al., 2023, see §5). This paper revisits and extends the view of GFlowNets as diversity-seeking learners in MDPs, enabling the training of robust agents in stochastic environments and two-player adversarial games.

The main application of GFlowNets to date has been generation of structured objects $x \in \mathcal{X}$ – where $\mathcal{X}$ is the set of terminal states of an episodic MDP – given a reward function $R : \mathcal{X} \to \mathbb{R}_{>0}$ interpreted as an unnormalized density. The generation of $x$ follows a trajectory $s_0 \to s_1 \to \cdots \to s_n = x$, representing iterative construction or refinement (*e.g.*, building a graph by adding one edge at a time). These settings assume that applying an action to a partially constructed object $s_i$ deterministically yields the object $s_{i+1}$. It is natural to attempt to generalize the GFlowNet framework to stochastic environments, in which a given sequence of actions does not always produce the same final state.

However, the common notions of GFlowNets must be modified to recover a useful stochastic generalization. An existing attempt (Pan et al., 2023) proposes to treat stochastic transitions in the environment as actions of the GFlowNet drawn from a fixed policy. One of the starting points for this paper is that this previous formulation sacrifices several desirable theoretical properties, inducing poor sampling performance in many practical settings. We propose an alternative notion of *expected flow networks* (EFlowNets), which provably do not suffer from the previous limitations (Fig. 1a).

As EFlowNets can perform inference via stochastic control in a fixed environment, they can be used to learn robust strategies against a stochastic opponent in a two-player game. We further define an *adversarial flow network* (AFlowNet) as a collection of EFlowNet players, each with its own reward function, taking actions in a shared environment. We show the existence and uniqueness of a joint optimum of the players' respective objectives. This stable point can be characterized via a probabilistic notion of game-theoretic equilibrium. We perform additional theoretical analysis and develop efficient training objectives for two-player zero-sum games.

---

[*]Equal contribution. [◇]CIFAR Senior Fellow. [†]CIFAR AI Chair.

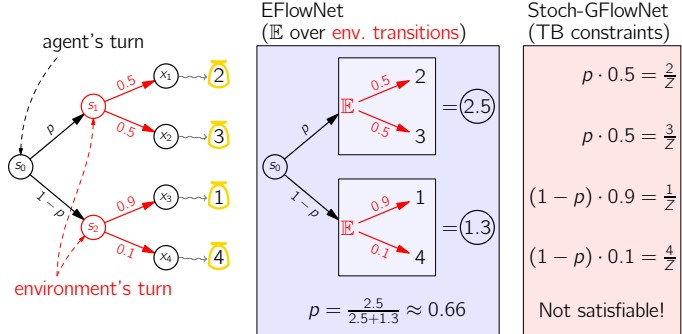
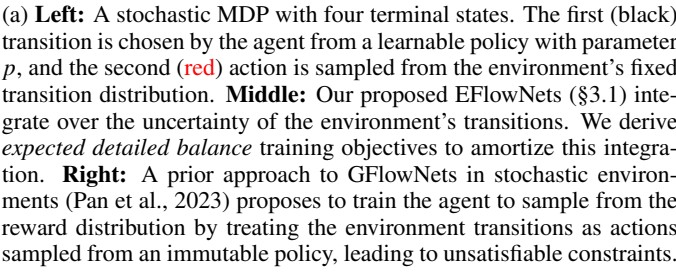
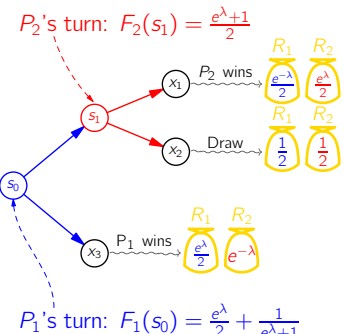

(a) **Left:** A stochastic MDP with four terminal states. The first (black) transition is chosen by the agent from a learnable policy with parameter $p$, and the second (red) action is sampled from the environment's fixed transition distribution. **Middle:** Our proposed EFlowNets (§3.1) integrate over the uncertainty of the environment's transitions. We derive *expected detailed balance* training objectives to amortize this integration. **Right:** A prior approach to GFlowNets in stochastic environments (Pan et al., 2023) proposes to train the agent to sample from the reward distribution by treating the environment transitions as actions sampled from an immutable policy, leading to unsatisfiable constraints.

(b) Optimizing two EFlowNets that play against each other yields robust game-playing agents. Here, a branch-adjusted AFlowNet (§3.3) for a two-player zero-sum game is shown: each player receives a reward of $e^\lambda$ for a win, 1 for a draw, and $e^{-\lambda}$ for a loss, adjusted appropriately by branching factors.

Figure 1: We extend GFlowNets to stochastic environments (a) and games (b).

The contributions of this work are as follows:

(1) We propose *expected flow networks* (EFlowNets, §3.1), a class of sequential sampling models and learning objectives generalizing GFlowNets on tree-structured state spaces. We demonstrate theoretically and experimentally the advantages of the EFlowNet formulation over past attempts to generalize GFlowNets to stochastic environments (§3.1, §4.1).

(2) We define *adversarial flow networks* (AFlowNets, §3.2) for two-player games and prove the existence and uniqueness of equilibrium solutions. In Proposition 4 we exploit the zero-sum structure of two-player games to get a novel trajectory balance (TB) loss for AFlowNets. We believe this new loss is a major algorithmic novelty. We conduct extensive experiments on two-player games showing that AFlowNets are able to learn robust policies via self-play (§4.2).

(3) We connect GFlowNets, EFlowNets, and AFlowNets to models of imperfect agents in psychology and behavioral economics (§A).

## 2 BACKGROUND

### 2.1 GFLOWNETS IN TREE-SHAPED ENVIRONMENTS

We review GFlowNets in deterministic environments, mainly following the conventions from Malkin et al. (2022). (Notably, we assume that terminating states have no children and rewards are strictly positive.) In keeping with past work, we use the language of directed acyclic graphs (DAGs), rather than the equivalent language of deterministic MDPs. We state all results only for *tree-structured* state spaces, but note that they are special cases of results for general DAGs.

**Setting.** Let $G = (\mathcal{S}, \mathcal{A})$ be a directed tree, with finite sets of vertices (*states*) $\mathcal{S}$ and edges (*actions*) $\mathcal{A} \subset \mathcal{S} \times \mathcal{S}$, oriented away from the root (*initial state*) $s_0 \in \mathcal{S}$. Denote by $\text{Ch}(s)$ the set of children of a state $s$ and by $\text{Pa}(s)$ the parent of $s$, which exists unless $s = s_0$. The set of childless (*terminal*) states is denoted $\mathcal{X}$. A *complete trajectory* is a sequence $\tau = (s_0 \to s_1 \to \cdots \to s_n)$, where $s_n \in \mathcal{X}$ and each $s_i \to s_{i+1}$ is an action. The set of complete trajectories is denoted $\mathcal{T}$.

A *(forward) policy* is a collection of distributions $P_F(\cdot \mid s)$ over $\text{Ch}(s)$ for every $s \in \mathcal{S} \setminus \mathcal{X}$. A policy induces a distribution over $\mathcal{T}$, with $P_F(s_0 \to s_1 \to \cdots \to s_n) = \prod_{i=1}^n P_F(s_i \mid s_{i-1})$. This in turn induces a *terminating distribution* $P_F^\top$ over $\mathcal{X}$, defined as the marginal distribution over the final state of a trajectory $\tau \sim P_F(\tau)$. One can sample $x \sim P_F^\top(x)$ by running a chain starting at $s_0$ and transitioning according to $P_F$ until a terminal state $x$ is reached.

A *reward function* is a function $R : \mathcal{X} \to \mathbb{R}_{>0}$. A policy $P_F$ is said to *sample proportionally to the reward* $R$ if $P_F^\top(x) \propto R(x)$, *i.e.*, $P_F^\top(x) = R(x)/Z$ for all $x \in \mathcal{X}$, where $Z = \sum_{x \in \mathcal{X}} R(x)$. Given a

reward function $R$, GFlowNet algorithms aim to produce a policy $P_F$ that samples proportionally to $R$. This is, in essence, a generative modeling problem, but the setting is close to maximum-entropy reinforcement learning (RL; Haarnoja et al., 2017): one is not given samples from the target density, as in typical generative modeling settings, but must rather *explore* the reward landscape through sequential sampling.

**FM and DB objectives.**   We review the two GFlowNet objectives of *flow matching* (FM; Bengio et al., 2021) and *detailed balance* (DB; Bengio et al., 2023) in tree-structured state spaces.

The FM objective optimizes a function $F : \mathcal{S} \rightarrow \mathbb{R}_{>0}$, called the *state flow*. The objective enforces a pair of constraints, which in tree-structured DAGs are

$$F(s) = \sum_{s' \in \text{Ch}(s)} F(s') \ \ \forall s \in \mathcal{S} \setminus \mathcal{X} \qquad \text{and} \qquad F(x) = R(x) \ \ \forall x \in \mathcal{X}. \tag{1}$$

Any edge flow $F$ induces a policy $P_F$, defined by $P_F(s' \mid s) = \frac{F(s')}{F(s)}$ for all $(s, s') \in \mathcal{A}$. If the flow satisfies the FM constraints (1), then it holds that $P_F$ samples proportionally to the reward $R$.

The DB objective avoids the explicit summation over children in (1) and jointly optimizes both $F$ and the policy $P_F$, replacing the first constraint by

$$F(s)P_F(s' \mid s) = F(s') \ \ \forall (s, s') \in \mathcal{A}. \tag{2}$$

This constraint implies the first constraint of (1), as $P_F$ sums to 1 over $s' \in \text{Ch}(s)$.

The function $F$ is typically parametrized as a neural network $F_\theta$ with parameters $\theta$, and (if using the DB objective) the policy $P_F$ as a network producing logits of $P_F(s' \mid s; \theta)$ given $s$ as input. The parameters $\theta$ are optimized to minimize some discrepancy between the left and right sides of (1) or (2). A typical choice is the squared log-ratio; for example, the FM objective at a state $s$ is

$$\mathcal{L}_{\text{FM}}(s) = \left( \log F_\theta(s) - \log \sum_{s' \in \text{Ch}(s)} F_\theta(s') \right)^2. \tag{3}$$

The choice of states $s$ at which this objective is evaluated and optimized is made by a *training policy* $\pi$. For example, $\pi$ could select the states $s$ seen in trajectories sampled from $P_F$ (*on-policy training*), but could also use off-policy exploration techniques, such as tempering, replay buffers, or Thompson sampling (Rector-Brooks et al., 2023). Because the objective can be simultaneously minimized to zero at all $s$ for a sufficiently expressive $F_\theta$, the global optimum of the objective is independent of the choice of training policy $\pi$, as long as $\pi$ has full support. This capacity for off-policy training without differentiating through the sampling procedure is a key advantage of GFlowNets over on-policy RL algorithms and over other hierarchical variational methods (Malkin et al., 2023).

**Connections with RL.**   In the case of tree-structured state spaces, GFlowNets are closely connected to entropy-regularized RL methods (soft Q-learning; Haarnoja et al., 2017): identifying the log-flow function with a value function, the FM/DB objectives are analogous to temporal difference learning (Sutton & Barto, 2018) and TB, along with its variant SubTB (Madan et al., 2023), to path consistency learning (Nachum et al., 2017). As a diversity-seeking agent, a GFlowNet can also be understood as way to train a quantal response agent; see §A for more discussion.

**How restrictive is the tree structure?**   Any environment that has a non-tree DAG structure – *i.e.*, where multiple trajectories may lead to the same state – can be converted to a tree-structured environment by augmenting each state with the history (the trajectory followed to reach the state). This implicitly multiplies the reward of each terminal state by the number of trajectories that lead to it (while keeping the optimal policy independent of the history). This alternative way to reward the state may be desired in some applications (*e.g.*, zero-sum games) for which the path to the solution matters as much as the outcome.

## 2.2   PAST APPROACHES TO GFLOWNETS IN STOCHASTIC ENVIRONMENTS

Pan et al. (2023) propose a generalization of GFlowNets to stochastic environments (*i.e.*, where the state and choice of action nondeterministically yield the subsequent state), following an approach described in Bengio et al. (2023). We now review their formulation, which we refer to as 'stochastic GFlowNets', restating it in a suitable language to motivate our method.

In stochastic environments, every state $s$ is associated with a set of possible actions $\mathcal{A}_s$, and the environment provides a stochastic transition function – a distribution $P_{\text{env}}(s' \mid s, a)$, understood as the likelihood of arriving in state $s'$ when taking action $a$ at state $s$. In stochastic GFlowNets, the state space $\mathcal{S}$ is augmented with a collection of *hypothetical states* $(s, a)$ for $s \in \mathcal{S} \setminus \mathcal{X}$ and $a \in \mathcal{A}_s$. The augmented DAG $G$ contains two kinds of edges:

- Edges $s \rightarrow (s, a)$ for each $s \in \mathcal{S}$ and $a \in \mathcal{A}_s$, which we call *agent edges*;
- Edges $(s, a) \rightarrow s'$ for each hypothetical $s'$ in the support of $P_{\text{env}}(\cdot \mid s, a)$, $a \in \mathcal{A}_s$, and $s' \in \text{Ch}(s)$, which we call *environment edges*.

Stochastic GFlowNets directly apply the training algorithms applicable to deterministic GFlowNets (*e.g.*, DB) to the augmented DAG $G$, with the only modification being that the forward policy $P_F$ is free to be learned only on agent edges, while on environment edges it is fixed to the transition function. Formally, for all $s \in \mathcal{S} \setminus \mathcal{X}$ and $a \in \mathcal{A}_s$, one learns $P_F((s, a) \mid s)$ (denoted $P_F(a \mid s)$ for short), while $P_F(s' \mid (s, a))$ is fixed to $P_{\text{env}}(s' \mid s, a)$.

The environment policy $P_{\text{env}}$, which appears in the loss, may be assumed to be known, but may also be approximated using a neural network trained by a maximum-likelihood objective on observed environment transitions, jointly with the agent policy.

**Violated desiderata in stochastic GFlowNets.** By construction, if a stochastic GFlowNet satisfies the DB constraints, then the policy $P_F$ samples proportionally to the reward $R$. In this way, stochastic GFlowNets are a minimal modification of GFlowNets that can function in stochastic environments. However, there exist stochastic environments and reward functions for which *no* stochastic GFlowNet policy $P_F(a \mid s)$ can satisfy the constraints (Fig. 1a). Two consequences of this are the impossibility of minimizing the loss to zero for all transitions, even for a perfectly expressive policy model, and the resulting dependence of the global optimum on the choice of training policy $\pi$. Thus stochastic GFlowNets satisfy D0, but not D1 (as noted by Bengio et al. (2023)) and D2 below.

The generalization of GFlowNet constraints and objectives to stochastic environments that we propose satisfies the following desiderata:

D0. If the environment's transition function $P_{\text{env}}$ is deterministic, one should recover deterministic GFlowNet constraints and objectives.
D1. *Satisfiability:* A perfectly expressive model should be able to minimize the generalized FM/DB losses to 0 for all states/actions in the DAG simultaneously. Consequently, the set of global optima of the loss should not depend on the choice of full-support training policy.
D2. *Uniqueness:* If $G$ is a tree, then the global optimum of the loss should be unique.
D3. *Equilibrium:* In a game where two GFlowNet agents alternate actions, there should be a unique pair of policies for the two players such that each policy is optimal for its respective loss.

As noted in §2.1, deterministic GFlowNets satisfy D1 and D2. D0 is a common-sense property, as deterministic environments are special cases of stochastic environments. D1 (satisfiability) is essential for off-policy training, while D2 (uniqueness) is desirable in game-playing agents. The meaning of D3 will be detailed in §3.2.

## 3 Method: Expected and adversarial flow networks

### 3.1 Expected flow networks

In this section, we define expected flow networks (EFlowNets) on tree-structured spaces, which encompasses the problems we study, in particular, two-player games with memory. We then show that EFlowNets satisfy the desiderata D0–D2 above.

Expected flow networks (EFlowNets) assume the following are given:

- A tree $G = (\mathcal{S}, \mathcal{A})$, with initial state $s_0$ and set of terminal states $\mathcal{X}$, and a reward function $R : \mathcal{X} \rightarrow \mathbb{R}_{>0}$.
- A partition of the nonterminal states into two disjoint sets, $\mathcal{S} \setminus \mathcal{X} = \mathcal{S}_{\text{agent}} \sqcup \mathcal{S}_{\text{env}}$, called the *agent states* and *environment states*, respectively.
- A distribution $P_{\text{env}}(\cdot \mid s)$ over the children of every environment state $s \in \mathcal{S}_{\text{env}}$.

Observe that if $\mathcal{S}_{\text{env}} = \emptyset$, then the input data for an EFlowNet is the same as the input data for a GFlowNet on a tree-structured space. This setting also generalizes that of stochastic GFlowNets

in §2.2, where all transitions link agent states $s$ to environment states $(s, a)$ – called 'hypothetical states' by Pan et al. (2023) – or vice versa.

An *agent policy* is a collection of distributions $P_{\text{agent}}(\cdot \mid s)$ over the children of every agent state $s \in \mathcal{S}_{\text{agent}}$. Together, $P_{\text{agent}}$ and $P_{\text{env}}$ determine a forward policy on $G$. We define the *expected detailed balance* (EDB) constraints relating $P_{\text{agent}}$, $P_{\text{env}}$, and a state flow function $F : \mathcal{S} \to \mathbb{R}_{>0}$:

$$F(s)P_{\text{agent}}(s' \mid s) = F(s') \qquad\qquad \forall s \in \mathcal{S}_{\text{agent}}, s' \in \text{Ch}(s), \qquad (4)$$

$$F(s) = \mathbb{E}_{s' \sim P_{\text{env}}(s'|s)} F(s') \qquad\qquad \forall s \in \mathcal{S}_{\text{env}}, \qquad (5)$$

$$F(x) = R(x) \qquad\qquad \forall x \in \mathcal{X}. \qquad (6)$$

These constraints satisfy the desiderata D0–D2, as summarized in the following proposition.

**Proposition 1.** *There exists a unique pair of state flow function $F$ and agent policy $P_{\text{agent}}$ satisfying constraints (4), (5), and (6). If $\mathcal{S}_{\text{env}} = \emptyset$, then this pair satisfies the detailed balance constraints (2).*

EFlowNets marginalize over the uncertainty of the environment's transitions: they aim to sample each action in proportion to the *expected* total reward available if the action is taken (see Prop. 5). A connection between EFlowNets and Luce quantal response agents is made in §A.

**Training EFlowNets: From constraints to losses.**   Just as in deterministic environments, when training EFlowNets, we parametrize the state flow and agent policy as neural networks $F_\theta$ and $P_{\text{agent}}(\cdot \mid \cdot; \theta)$. The EDB constraints can be turned into squared log-ratio losses in the same manner that the FM constraint (1) is converted into the loss (3) and optimized by gradient descent.

In problems where the number of environment transitions is large and computing the expectation on the right side of (5) is costly, it may be replaced by an alternative constraint by introducing a distribution $Q(s' \mid s) = \frac{F(s')P_{\text{env}}(s'|s)}{F(s)}$. This quantity sums to 1 over the $s'$ if and only if (5) is satisfied. Thus (5) is equivalent to the following constraint on $Q$:

$$F(s)Q(s' \mid s) = F(s')P_{\text{env}}(s' \mid s). \qquad (7)$$

Enforcing this constraint requires learning an additional distribution $Q(s' \mid s; \theta)$, but does not require summation over children. The conversion from (5) to (7) resembles that from FM (1) to DB (2).

Just like deterministic GFlowNets, the globally optimal agent policy in an EFlowNet is unique and does not depend on the distribution of states at which the objectives are optimized, as long as it has full support. However, the choice of training policy can be an important hyperparameter that can affect the rate of convergence and the local minimum reached in a function approximation setting. We describe the choices we make in the experiment sections below.

Just like in stochastic GFlowNets (§2.2), the environment policy $P_{\text{env}}$ can be either assumed to be known or learned, jointly with the policy, from observations of the environment's transitions.

## 3.2 Adversarial flow networks

We now consider the application of EFlowNets to multiagent settings. Although our experiments are in the domain of two-player games, we define adversarial flow networks (AFlowNets) in their full generality, with $n$ agents. AFlowNets with $n$ agents, or players, depend on the following information:

- A tree $G = (\mathcal{S}, \mathcal{A})$, with initial state $s_0$ and set of terminal states $\mathcal{X}$, and a collection of reward functions $R_1, \dots, R_n : \mathcal{X} \to \mathbb{R}_{>0}$.
- A partition of the nonterminal states into disjoint sets, $\mathcal{S} \setminus \mathcal{X} = \mathcal{S}_1 \sqcup \cdots \sqcup \mathcal{S}_n$.

This data defines a fully observed sequential game, where $s \in \mathcal{S}_i$ means that player $i$ is to play at $s$. An *agent policy* for player $i$ is a collection of distributions $P_i(\cdot \mid s)$ over $\text{Ch}(s)$ for every $s \in \mathcal{S}_i$.

The input data for an AFlowNet also defines a collection of EFlowNets, one for each player $i$. The EFlowNet $\mathcal{E}_i$ for player $i$ has the same underlying graph $G$, with $\mathcal{S}_{\text{agent}} = \mathcal{S}_i$ and $\mathcal{S}_{\text{env}} = \bigsqcup_{j \neq i} \mathcal{S}_j = \mathcal{S} \setminus (\mathcal{X} \cup \mathcal{S}_i)$, and reward function $R_i$. That is, each player is viewed as an agent in an EFlowNet whose 'environment' is given by the other players' policies. (We also remark that the case $n = 1$ recovers a regular (deterministic) GFlowNet.)

The policy $P_i$ of player $i$ can be optimized using the EFlowNet training objective given fixed values of the other players' policies $P_j$ ($j \neq i$), and by Proposition 1, there is a unique global optimum for $P_i$. However, remarkably, there exists a unique collection of policies $P_1, \dots, P_n$ such that each $P_i$ that *jointly* satisfy the EFlowNet constraints for each player.

**Proposition 2.** *There exist unique agent policies $P_1, \ldots, P_n$ and state flow functions $F_1, \ldots, F_n$ : $\mathcal{S} \to \mathbb{R}_{>0}$ such that $P_i$ and $F_i$ satisfy the EDB constraints with respect to the EFlowNet $\mathcal{E}_i$ for all i.*

We also have a characterization of the joint optimum in the case of two-agent AFlowNets:

**Proposition 3.** *Suppose that in a 2-player AFlowNet, the agent policies $P_1, P_2$ and state flow functions $F_1, F_2$ are jointly optimal in the sense of Prop. 2. Then the function $F(s) = F_1(s)F_2(s)$ is a flow on G, i.e., satisfies the FM constraint (1), with respect to the reward $R(x) = R_1(x)R_2(x)$.*

### 3.3 Branch-adjusted AFlowNets for two-player zero-sum games.

An important application of AFlowNets is two-player zero-sum games. We will now describe a way to turn the game outcomes into rewards that allows a simpler and more efficient training objective.

Specifically, we consider a two-player, complete-information game with tree-shaped state space $G$, in which player 1 moves first (*i.e.*, $s_0 \in \mathcal{S}_1$) and the players alternate moves, *i.e.*, every complete trajectory $s_0 \to s_1 \to \cdots \to s_n = x$ has $s_i \in \mathcal{S}_1$ if and only if $i$ is even. We assume that the game ends in a win for player 1, a win for player 2, or a draw. A naïve way to define the rewards for players 1 and 2 is the following, which ensures the log-rewards sum to zero at every terminal state:

$$R_i^\circ(x) = \begin{cases} e^\lambda & \text{if player } i \text{ wins,} \\ 1 & \text{if the game ends in a draw,} \\ e^{-\lambda} & \text{if player } i \text{ loses.} \end{cases} \tag{8}$$

However, we find that AFlowNets trained with this reward often exhibit suboptimal behaviour in complex games: the agent may avoid a move that leads directly to a winning terminal state (high reward) in favour of a move with a large downstream subtree. We therefore define an alternative reward function that favours shorter winning trajectories. If $s_0 \to s_1 \to \cdots \to s_n = x$ is the trajectory leading to $x$, then the branch-adjusted reward for player $i$ is defined as

$$R_i(x) = \frac{R_i^\circ(x)}{B_i(x)}, \qquad B_i(x) := \prod_{k: s_k \in \mathcal{S}_k} |\text{Ch}(s_k)|. \tag{9}$$

That is, $B_i(x)$ is the product of the branching factors (numbers of children) of the states $s_k$ on the trajectory at which player $i$ is to move.[1] Besides delivering a higher reward for less selective trajectories and being empirically essential for good game-playing performance, this correction is necessary to derive the simplified objective below, which critically uses $R_1^\circ(x)R_2^\circ(x) = 1$.

**A 'trajectory balance' for branch-adjusted AFlowNets.** A limitation of objectives such as FM, DB, and their EFlowNet and AFlowNet generalizations are their slow credit assignment (propagation of a reward signal) over long trajectories, which results from these losses being local to a state or transition. This limitation motivated the trajectory balance (TB) loss for GFlowNets (Malkin et al., 2022), which delivers a gradient update to all policy probabilities along a complete trajectory.

While the GFlowNet TB objective does not appear to generalize to expected flow networks, we derive an objective of a TB-like flavour for branch-adjusted AFlowNets.

**Proposition 4.** *In a 2-player AFlowNet with alternating moves satisfying $R_1^\circ(x)R_2^\circ(x) = 1$:*

(a) *Suppose that the agent policies $P_1, P_2$ and state flow functions $F_1, F_2$ are jointly optimal in the sense of Prop. 2. Then there exists a scalar Z, independent of x, such that for every complete trajectory $s_0 \to s_1 \to \cdots \to s_n = x$,*

$$Z \prod_{i: s_i \in \mathcal{S}_1} P_1(s_{i+1} \mid s_i) = R_1(x)B_2(x) \prod_{i: s_i \in \mathcal{S}_2} P_2(s_{i+1} \mid s_i). \tag{10}$$

(b) *Conversely, if the constraint (10) holds for some constant Z and policies $P_1$ and $P_2$, then $P_1$ and $P_2$ are the jointly optimal AFlowNet policies.*

The constraint (10) can be converted into a training objective $\mathcal{L}_{\text{TB}}$ – the squared log-ratio between the left and right sides – and optimized with respect to the policy parameters and the scalar $Z$ (parametrized through $\log Z$ for stability) for complete trajectories (game simulations) sampled from a training policy. Prop. 2 and Prop. 4(a) guarantee that the constraints are satisfiable, while Prop. 4(b) guarantees that the policies satisfying the constraints are unique.

---

[1] While the branching factor appears large, it is counteracted by the fact that, on an AFlowNet agent's turn, its child flows are summed, *e.g.*, $F_1(s) = \sum_{s' \in \text{Ch}(s)} F_1(s')$ if $s \in \mathcal{S}_1$.

**Training AFlowNets.** AFlowNets are trained by optimizing the EFlowNet objectives of each agent independently. The states at which the objectives are optimized are chosen by a training policy, which may either sample the agents' policies to produce training trajectories (*on-policy self-play*) or use off-policy exploration. The joint global optimum, where all agents optimize their losses to 0, is unique and independent of the training policy due to Prop. 2. See Alg. 1.

A significant benefit of AFlowNets over methods like Silver et al. (2018) is that they do not require expensive rollout procedures (*i.e.*, MCTS) to generate games. MCTS performs a number of simulations – each of which requires a forward pass – for every state in a game. AFlowNets, on the other hand, only require a single forward pass per state. Consequently, AFlowNets can be trained on more games given a similar computational budget.

## 4 EXPERIMENTS

We conduct experiments to investigate whether EFlowNets can effectively learn in stochastic environments compared to related methods (§4.1) and whether AFlowNets are effective learners of adversarial gameplay, as measured by their performance against contemporary approaches (§4.2).

### 4.1 GENERATIVE MODELING IN STOCHASTIC ENVIRONMENTS

We evaluate EFlowNets in a protein design task from Jain et al. (2022). The GFlowNet policy autoregressively generates an 8-symbol DNA sequence $x$ and receives a reward of $R(x) = f(x)^\beta$, where $f(x)$ is a proxy model estimating binding affinity to a target protein and $\beta$ is a hyperparameter controlling the reward distribution's entropy. In Pan et al. (2023), the problem was made stochastic by letting the environment replace the symbol appended by the policy to the right of a partial sequence with a uniformly random symbol with probability $\alpha$. Thus $\alpha = 0$ gives a deterministic environment and $\alpha = 1$ a fully stochastic environment, where the policy's actions have no effect.

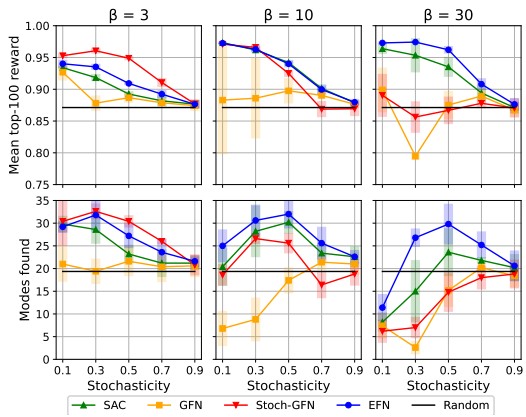

Figure 2: Results on the TFBind task (five seeds per setting). EFlowNets tend to find more diverse high-reward states, especially when the reward is peaky and environment stochasticity is high, making the Stoch-GFN constraints unsatisfiable.

We extend the published code of Pan et al. (2023) with an implementation of the EFlowNet objective. Besides the stochastic GFlowNet (Stoch-GFN) formulation from §2.2, we compare with the two strongest baselines considered in that work: a "naïve" GFlowNet that ignores the environment's transitions (GFN), and a discrete soft actor-critic (SAC; Haarnoja et al., 2018). We use the hyperparameters from the existing implementation for all methods (except SAC, which we reimplemented because code was not available) and report the same primary metrics: the mean reward of the top-100 sequences among 2048 sampled from a trained model and the number of diverse modes found, as measured by the sphere exclusion algorithm from Jain et al. (2022). A model of the environment's transition distribution is learned, consistent with Pan et al. (2023).

The results and error ranges, with different values of the stochasticity $\alpha$ and reward exponent $\beta$, are shown in Fig. 2. When the reward is peaky (larger $\beta$), EFlowNets outperform other algorithms in both diversity and top-100 reward. This is consistent with situations such as those in Fig. 1a, where the Stoch-GFN constraints are unsatisfiable, being more common when the reward is peaky, as the environment's random actions place smoothness constraints on the sampling distribution. Of note, our implementation of SAC performs better than what is reported in Pan et al. (2023) and often better than Stoch-GFN, which was previously considered only with the flat-reward setting of $\beta = 3$.

### 4.2 ADVERSARIAL GAMES

The game-playing capabilities of AFlowNets are evaluated in 2-player games. Rewards are modeled as described in §3.3, and the AFlowNets trained with rewards defined by (8) with a given $\lambda$ are denoted AFlowNet$_\lambda$. We evaluate how the efficacy of an agent changes with various values of $\lambda$.

---

**Algorithm 1:** Branch-adjusted AFlowNet Training

**Data:** $\lambda$, batch size $n$, number of trajectories $K$, number of steps $L$, buffer capacity $M$,
model and training hyperparameters
Initialize AFlowNet policies $P_1, P_2, \log Z$, and replay buffer $B$ with capacity $M$
**for** $i = 1$ **to** $N$ **do**
    Generate $K$ trajectories (sampling from AFlowNet's policies $P_1, P_2$)
    Add trajectories to $B$ // $B$ is a FIFO queue
    **for** $i = 1$ **to** $L$ **do**
        $\{\tau_j\}_{j=1}^n \leftarrow$ Sample randomly from $B$
        $\mathcal{L} \leftarrow \frac{1}{n} \sum_{j=1}^n \mathcal{L}_{\text{TB}}(\log Z, P_1, P_2, \tau_j, \lambda)$
        Gradient update on $\nabla \mathcal{L}$ with respect to $\log Z$ and policy parameters

---

AFlowNets are trained using the TB loss to play tic-tac-toe and Connect-4. For each, we run a tournament against an open-source AlphaZero implementation (Thakoor et al., 2016) and a uniform random agent. For tic-tac-toe we also include a tree-search agent which uses AlphaZero's value function, alpha-beta pruning, and a search depth of 3. To compare the agents, we compute their Elo over the course of training using BayesElo (Coulom, 2008; Diaz & Bück-Kaeffer, 2023). The training procedure is outlined in Alg. 1 and details are in §D.

Figure 3 (left) illustrates the Elo of the agents in tic-tac-toe over training steps and time. It is clear that AFlowNets quickly achieve a competitive Elo with AlphaZero. It is worth noting that AFlowNet$_2$ and AFlowNet$_{15}$ achieved and Elo of $334.8 \pm 15.5$ and $231.1 \pm 91.3$, whereas AFlowNet$_{10}$ achieved an Elo of $338.4 \pm 14.1$. As such, it appears that $\lambda$ has a diminishing return on game performance in tic-tac-toe.

The parameter $\lambda$ has a large effect on the performance of AFlowNets in Connect-4 (cf. Fig. 3 (right)). The AFlowNets with $\lambda \in \{10, 15\}$ achieve the highest Elo of all tested agents. AFlowNets win almost every game against AlphaZero and achieve an Elo score roughly 800 points higher. Additional tournament results and further analysis are available in §E.

As in Prasad et al. (2018), we take advantage of the fact that Connect-4 is solved (Allis, 1988) to obtain perfect values for arbitrary positions. We compare the moves selected by the AFlowNet with the values computed by a perfect Connect-4 solver (Pons (2019)) over the course of training. Fig. 4 shows an evaluation of the AFlowNet's performance using this metric (and baseline values for a random uniform agent). The AFlowNets learn to play optimal moves in $> 80\%$ of board states after 3 hours of training on one RTX8000 GPU.

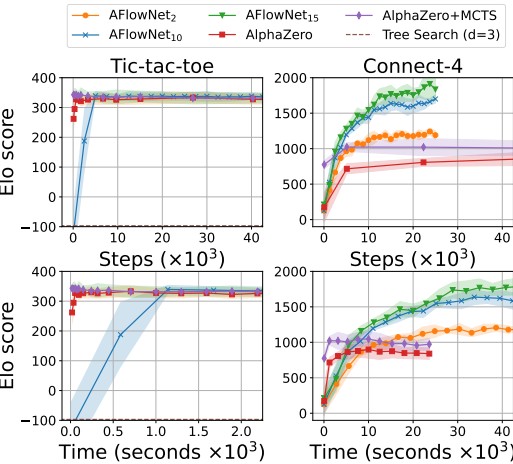

Figure 3: Elo as a function of training steps and training time. As a convention, random uniform baseline agents represent an Elo of 0. AFlowNets achieve similar Elo to AlphaZero in tic-tac-toe and AFlowNets quickly learn to outperform AlphaZero in Connect-4.

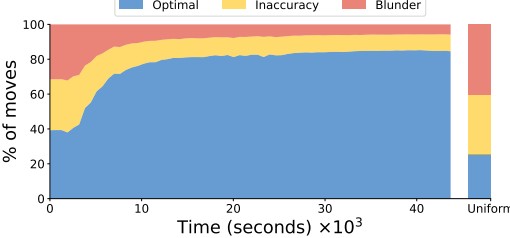

Figure 4: Move quality for the AFlowNet (for a set of 10240 randomly generated Connect-4 boards) over the course of training. An *optimal* move leads to the quickest win or slowest loss. An *inaccuracy* is a non-optimal move that has the same sign as the optimal move (*e.g.*, leading to a win but not as quickly). A *blunder* leads from a winning state to either a drawing or losing state.

Table 1: Summary of the main differences between AFlowNet and AlphaZero training.

|  | AFlowNet | AlphaZero |
| --- | --- | --- |
| Action sampling | single forward pass | MCTS |
| Objective input | complete trajectory | single state |
| States per optim. step | batch size × traj. length | batch size |

**Differences between AFlowNets and AlphaZero methodologies.** Distinctions exist between our approach and AlphaZero that make direct comparisons between the methods challenging, summarized in Table 1. Most importantly, the batch-adjusted AFlowNet objective depends on an entire game simulation, while the AFlowNet value function updates are performed at individual states. In addition, the game simulations in AFlowNets are obtained using a single policy rollout, without Monte Carlo tree search. Thus, generation of training data with AFlowNets is faster than with AlphaZero, assuming the base model architectures are of a similar scale.

**Demo.** We invite the reader to play Connect-4 anonymously against a pretrained AFlowNet agent at the following URL: `https://bit.ly/demoafn`.

## 5 RELATED WORK

**Stochasticity in GFlowNets.** GFlowNets have been used as diversity-seeking samplers in various settings with deterministic environment transitions. In particular, they have been interpreted as hierarchical variational inference algorithms (Malkin et al., 2023; Zimmermann et al., 2023) and correspondingly applied to modeling of Bayesian posteriors (Deleu et al., 2022; van Krieken et al., 2022; Hu et al., 2023). GFlowNets can be trained with stochastic *rewards* (Bengio et al., 2023), and Deleu et al. (2022; 2023); Liu et al. (2022) take advantage of this property to train samplers of Bayesian posteriors using small batches of observed data. Zhang et al. (2023b) proposed to match the uncertainty in a stochastic reward in a manner resembling distributional RL (Bellemare et al., 2017); however, the stochasticity is introduced only at terminal states, while we consider stochasticity in intermediate transitions. The stochastic modelling of Pan et al. (2023), as we have argued, is insufficient to capture desired sampling behaviours in the problems we consider.

**RL in stochastic environments and games.** Learning in an environment with stochastic transition dynamics and against adversaries has long been a task of RL (Sutton & Barto, 2018). While AlphaZero (Silver et al., 2018) has achieved state-of-the-art performance in chess, Shogi, and Go, it does not explicitly model stochastic transition dynamics. Stochastic MuZero (Antonoglou et al., 2022) is a model-based stochastic planning method that learns a model of the environment and a policy at the same time, allowing it to perform well in a variety of stochastic games. Both AlphaZero and MuZero use Monte Carlo tree search for policy and value estimation (Silver et al., 2018; Antonoglou et al., 2022; Schrittwieser et al., 2020). Our EFlowNets bear similarities to a recently introduced approach that integrates over environment uncertainty in RL (Yang et al., 2023).

## 6 DISCUSSION

This paper extends GFlowNets to stochastic and adversarial environments. Expected flow networks learn in settings with stochastic transitions while maintaining desirable convergence properties, and adversarial flow networks pit EFlowNets against themselves in self-play. We successfully applied these algorithms to a stochastic generative modeling problem and to two real-world zero-sum games.

In future work, we intend to scale these methods to larger game spaces (*e.g.*, chess and Go). Such scaling is likely to require algorithmic improvements to address the limitations of our method. While we derived an efficient 'trajectory balance' for branch-adjusted AFlowNets, trajectory-level objectives suffer from high variance for long trajectories and have a high memory cost. Although these limitations did not surface in our experiments, it would be interesting to consider interpolations between expected DB and TB, akin to subtrajectory balance for GFlowNets (Madan et al., 2023).

Other future work can consider generalizations to incomplete-information games and cooperative multi-agent settings. For games with continuous action spaces, one can analyze the continuous-time and infinite-agent (mean-field) limits. Beyond games, GFlowNets and EFlowNets, in which node flows are computed by aggregation over children – either summation (4) or expectation (5) – may fall into a more general class of probabilistic flow networks that encompass a range of samplers trainable by local consistency objectives, reminiscent of the manner in which the language of circuits unifies probabilistic models with tractable inference (Choi et al., 2020; Vergari et al., 2021).

ACKNOWLEDGMENTS

The authors thank Moksh Jain for help with baseline code for the experiments in §4.1 and Manfred Diaz for help with the Elo computations in §4.2. We also thank Quentin Bertrand and Juan Duque for their comments on a draft of the paper.

YB acknowledges funding from CIFAR, NSERC, Intel, and Samsung.

GG acknowledges funding from CIFAR.

The research was enabled in part by computational resources provided by the Digital Research Alliance of Canada (`https://alliancecan.ca`), Mila (`https://mila.quebec`), and NVIDIA.

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

## A   GFLOWNETS AS QUANTAL RESPONSE AGENTS

**A GFlowNet policy as a Luce agent.**   GFlowNets in tree-structured spaces are closely related to probabilistic models of imperfect agent behaviour in game theory known as *quantal response* agents (McKelvey & Palfrey, 1996; 1995; Goeree et al., 2020). A particular kind of quantal response agent uses the Luce ratio rule (Luce, 1959) to sample strategies in proportion to their expected payoff. GFlowNet trajectories $\tau$ can be seen as (pure) strategies of an agent in a one-player game, and the reward $R(x)$ as the payoff for a trajectory $\tau$ leading to $x \in \mathcal{X}$. A GFlowNet that samples trajectories proportionally to the rewards of their last states, *i.e.*, $P_F(\tau = (s_0 \rightarrow s_1 \rightarrow \cdots \rightarrow s_n = x)) \propto R(x)$, is thus a Luce quantal response agent. On the level of individual actions at a given state $s$, the policy is that of a Luce agent that treats $F(s')$ – the total reward accessible from $s'$ – as the payoff for a transition $s \rightarrow s'$.

**EFlowNets learn a marginalized quantal response policy.**   Next, we show that EFlowNets are Bayesian (model-averaging) analogues of Luce agents that marginalize out the uncertainty of the environment's transitions.

Define a *(pure) environment strategy* as an induced subgraph $G_{\text{env}}$ of $G$ whose vertex set $V(G_{\text{env}}) \subseteq \mathcal{S}$ has the following properties:

- If $s \in V(G_{\text{env}})$ and $s \neq s_0$, then the parent of $s$ is in $V(G_{\text{env}})$.
- If $s \in V(G_{\text{env}}) \cap \mathcal{S}_{\text{env}}$, then exactly one child of $s$ is in $V(G_{\text{env}})$.

- If $s \in V(G_{\text{env}}) \cap \mathcal{S}_{\text{agent}}$, then all children of $s$ are in $V(G_{\text{env}})$.

It is clear from the first property that any such $G_{\text{env}}$ is a tree. An environment strategy thus amounts to a predetermined choice of action at every state that can be reached if the environment takes the actions chosen by the strategy. The environment policy $P_{\text{env}}$ determines a distribution over environment strategies, where the child of each agent state $s$ in $G_{\text{agent}}$ is sampled from the policy independently for each $s$, *i.e.*,

$$P_{\text{env}}(G_{\text{env}}) = \prod_{\substack{s \in V(G_{\text{env}}) \cap \mathcal{S}_{\text{env}} \\ s' \in \text{Ch}(s) \cap V(G_{\text{env}})}} P_{\text{env}}(s' \mid s). \tag{11}$$

The environment strategy is a source of uncertainty for the agent. Any given $G_{\text{env}}$ is a tree that contains $s_0$ and some subset $\mathcal{X}_{G_{\text{env}}}$ of the terminal states. Viewing $G_{\text{env}}$ as a (deterministic) GFlowNet in the sense of §2.1, there is a unique policy $P_F^{G_{\text{env}}}$, and corresponding state flow $F^{G_{\text{env}}}$ on $G_{\text{env}}$ that samples proportionally to the reward $R$ restricted to $\mathcal{X}_{G_{\text{env}}}$.

The following proposition shows that the optimal EFlowNet policy averages the stepwise utilities (*i.e.*, total accessible rewards) of deterministic-environment GFlowNets $P_F^{G_{\text{env}}}$ weighted by their likelihood under $P_{\text{env}}$.

**Proposition 5.** *Suppose that $P_{\text{agent}}$ satisfies the EDB constraints. Then, for any $s \in \mathcal{S}_{\text{agent}}$ and $s' \in \text{Ch}(s)$,*

$$P_{\text{agent}}(s' \mid s) \propto \mathbb{E}_{G_{\text{env}} \sim P_{\text{env}}} \left[ F^{G_{\text{env}}}(s') \mid s \in V(G_{\text{env}}) \right],$$

*where the expectation is taken over the distribution over strategies determined by $P_{\text{env}}$, restricted to the strategies that contain the state $s$.*

**AFlowNets and agent quantal response equilibrium.** In two-player games with unadjusted rewards $R_1^\circ, R_2^\circ$ and rewards $R_1$ and $R_2$ defined using the branching factor adjustment (9), we also have the following characterization of the optimal state flows:

$$B_i(s) F_i(s) = \mathbb{E}_{\substack{s=s_1 \to s_2 \to \ldots s_n = x \in \mathcal{X} \\ s_{k+1} \sim \mathcal{U}[\text{Ch}(s_k)] \text{ if } s_k \in \mathcal{S}_i \\ s_{k+1} \sim P_j(s_{k+1}|s_k) \text{ if } s_k \in \mathcal{S}_j \ (j \neq i)}} \left[ R_i^\circ(x) \right] \tag{12}$$

$$= \sum_{s=s_1 \to s_2 \to \ldots s_n = x \in \mathcal{X}} \left[ \prod_{k: s_k \in \mathcal{S}_i} \frac{1}{|\text{Ch}(s_k)|} \prod_{k: s_k \in \mathcal{S}_j, j \neq i} P_j(s_{k+1} \mid s_k) \right] R_i^\circ(x),$$

where the notation $B_i(s)$ is extended to nonterminal states $s$ using the same definition (9). This is easily derived by recursion from the EDB constraints and (9) in a similar way to the proof of Prop. 5. Because $P_i(s' \mid s) \propto F_i(s')$ for $s \in \mathcal{S}_i$, the expression (12) characterizes the policy $P_i$ via a form of *agent quantal response* (McKelvey & Palfrey, 1998), in which the action probability of an agent is proportional to its expected reward under future actions of the opponent (sampled from its policy) and the agent itself (here, sampled uniformly).

## B  PROOFS

**Proposition 1.** *There exists a unique pair of state flow function $F$ and agent policy $P_{\text{agent}}$ satisfying constraints (4), (5), and (6). If $\mathcal{S}_{\text{env}} = \emptyset$, then this pair satisfies the detailed balance constraints (2).*

*Proof of Proposition 1.* If the EDB constraints are satisfied, then $F$ satisfies a recurrence:

$$F(s) = \begin{cases} \sum_{s' \in \text{Ch}(s)} F(s') & s \in \mathcal{S}_{\text{agent}} \\ \mathbb{E}_{s' \sim P_{\text{env}}(s'|s)} F(s') & s \in \mathcal{S}_{\text{env}} \\ R(s) & s \in \mathcal{X} \end{cases}, \tag{13}$$

where the first case ($s \in \mathcal{S}_{\text{agent}}$) follows from summing (4) over $s'$. The uniqueness of $F(s)$ can easily be seen, *e.g.*, by induction on the length of the longest path from $s$ to a terminal state.

Because $R(x) > 0$ for all $x \in \mathcal{X}$, and the recurrence preserves the positivity (*i.e.*, $F(s') > 0$ for all $s' \in \text{Ch}(s)$ implies $F(s) > 0$), we have $F(s) > 0$ for all $s$. Therefore, one can recover the unique $P_{\text{agent}}$ that satisfies (4) jointly with $F$ via $P_{\text{agent}}(s' \mid s) = \frac{F(s')}{F(s)}$.

Finally, if $\mathcal{S}_{\text{env}} = \emptyset$, then constraint (5) is vacuous, and the remaining constraints exactly recover (2). □

**Proposition 2.** *There exist unique agent policies $P_1, \ldots, P_n$ and state flow functions $F_1, \ldots, F_n$ : $\mathcal{S} \to \mathbb{R}_{>0}$ such that $P_i$ and $F_i$ satisfy the EDB constraints with respect to the EFlowNet $\mathcal{E}_i$ for all $i$.*

*Proof of Proposition 2.* As in the proof of Proposition 1, we give a recurrence on the flows:

$$
F_i(s) = \begin{cases} \sum_{s' \in \mathrm{Ch}(s)} F_i(s') & s \in \mathcal{S}_i \\ \frac{\sum_{s' \in \mathrm{Ch}(s)} F_i(s') F_j(s')}{\sum_{s' \in \mathrm{Ch}(s)} F_j(s')} & s \in \mathcal{S}_j, j \neq i \\ R_i(s) & s \in \mathcal{X} \end{cases} \tag{14}
$$

This recurrence uniquely determines the state flows $F_i$ and therefore the policies $P_i$. It remains to see that if the flows satisfy (14), then each $F_i$ satisfies the recurrence (13). The cases $s \in \mathcal{S}_i$ and $s \in \mathcal{X}$ are clear. For the case $s \in \mathcal{S}_j, j \neq i$, observe that

$$
\frac{\sum_{s' \in \mathrm{Ch}(s)} F_i(s') F_j(s')}{\sum_{s' \in \mathrm{Ch}(s)} F_j(s')} = \sum_{s' \in \mathrm{Ch}(s)} \frac{F_i(s') F_j(s')}{\sum_{s'' \in \mathrm{Ch}(s)} F_j(s'')} = \sum_{s' \in \mathrm{Ch}(s)} F_i(s') P_j(s' \mid s) = \mathbb{E}_{s' \sim P_j(s' \mid s)} F_i(s'),
$$

which coincides with the second case of the recurrence (13). □

**Proposition 3.** *Suppose that in a 2-player AFlowNet, the agent policies $P_1, P_2$ and state flow functions $F_1, F_2$ are jointly optimal in the sense of Prop. 2. Then the function $F(s) = F_1(s)F_2(s)$ is a flow on $G$, i.e., satisfies the FM constraint (1), with respect to the reward $R(x) = R_1(x)R_2(x)$.*

*Proof of Proposition 3.* Because $F_1(x) = R_1(x)$ and $F_2(x) = R_2(x)$ for all $x \in \mathcal{X}$, we have $F(x) = R(x)$ for all $x \in \mathcal{X}$.

For $s \in \mathcal{S} \setminus \mathcal{X}$, we must show that $F(s) = \sum_{s' \in \mathrm{Ch}(s)} F(s')$. Suppose without loss of generality that $s \in \mathcal{S}_1$. Then

$$
F_1(s)F_2(s) = F_1(s) \mathbb{E}_{s' \sim P_1(s' \mid s)} F_2(s') = F_1(s) \sum_{s' \in \mathrm{Ch}(s)} \frac{F_1(s')}{F_1(s)} F_2(s') = \sum_{s' \in \mathrm{Ch}(s)} F_1(s') F_2(s'),
$$

which completes the proof. □

**Proposition 4.** *In a 2-player AFlowNet with alternating moves satisfying $R_1^\circ(x) R_2^\circ(x) = 1$:*

(a) *Suppose that the agent policies $P_1, P_2$ and state flow functions $F_1, F_2$ are jointly optimal in the sense of Prop. 2. Then there exists a scalar $Z$, independent of $x$, such that for every complete trajectory $s_0 \to s_1 \to \cdots \to s_n = x$,*

$$
Z \prod_{i:s_i \in \mathcal{S}_1} P_1(s_{i+1} \mid s_i) = R_1(x) B_2(x) \prod_{i:s_i \in \mathcal{S}_2} P_2(s_{i+1} \mid s_i). \tag{10}
$$

(b) *Conversely, if the constraint (10) holds for some constant $Z$ and policies $P_1$ and $P_2$, then $P_1$ and $P_2$ are the jointly optimal AFlowNet policies.*

*Proof of Proposition 4. Part (a).* We first extend the definition of $B_i$ to nonterminal states: if $s_0 \to s_1 \to \cdots \to s_m = s$ is any trajectory, define $B_i(s) := \prod_{0 \leq i < m : s_i \in \mathcal{S}_i} |\mathrm{Ch}(s_i)|$.

We claim that for all states $s$, $F_1(s)F_2(s) = \frac{1}{B_1(s)B_2(s)}$. This holds at terminal states $x$, since $F_1(x)F_2(x) = R_1(x)R_2(x) = \frac{R_1^\circ(x)R_2^\circ(x)}{B_1(x)B_2(x)} = \frac{1}{B_1(x)B_2(x)}$. By Prop. 3, $F_1(s)F_2(s)$ is a flow, so it suffices to show that $\frac{1}{B_1(s)B_2(s)}$ also satisfies (1) for $s \in \mathcal{X} \setminus \mathcal{S}$. Without loss of generality, suppose $s \in \mathcal{S}_1$ and let $s_0 \to \cdots \to s_i = s$ be the trajectory leading to $s$. Then

$$
\sum_{s' \in \mathrm{Ch}(s)} \frac{1}{B_1(s')B_2(s')} = \sum_{s' \in \mathrm{Ch}(s)} \frac{1}{B_1(s)|\mathrm{Ch}(s)| \cdot B_2(s)} = \frac{1}{B_1(s)B_2(s)},
$$

establishing the claim.

Returning to the proposition, rearranging factors and using the definition (9), it is necessary to show that

$$
\frac{R_1^\circ(x) B_2(x) \prod_{i:s_i \in \mathcal{S}_2} P_2(s_{i+1} \mid s_i)}{B_1(x) \prod_{i:s_i \in \mathcal{S}_1} P_1(s_{i+1} \mid s_i)}
$$

is independent of $x$. We have, using the above claim,

$$
\begin{aligned}
\frac{R_1(x)B_2(x)\prod_{i:s_i\in\mathcal{S}_2}P_2(s_{i+1}\mid s_i)}{\prod_{i:s_i\in\mathcal{S}_1}P_1(s_{i+1}\mid s_i)} &= \frac{R_1(x)\prod_{i:s_i\in\mathcal{S}_2}|\operatorname{Ch}(s_i)|\frac{F_2(s_{i+1})}{F_2(s_i)}}{\prod_{i:s_i\in\mathcal{S}_1}\frac{F_1(s_{i+1})}{F_1(s_i)}} \\
&= \frac{R_1(x)\prod_{i:s_i\in\mathcal{S}_2}|\operatorname{Ch}(s_i)|\frac{F_1(s_i)B_1(s_i)B_2(s_i)}{F_1(s_{i+1})B_1(s_{i+1})B_2(s_{i+1})}}{\prod_{i:s_i\in\mathcal{S}_1}\frac{F_1(s_{i+1})}{F_1(s_i)}} \\
&= \frac{R_1(x)\prod_{i:s_i\in\mathcal{S}_2}\frac{F_1(s_i)}{F_1(s_{i+1})}}{\prod_{i:s_i\in\mathcal{S}_1}\frac{F_1(s_{i+1})}{F_1(s_i)}} \\
&= R_1(x)\prod_{i<n\text{ even}}\frac{F_1(s_i)}{F_1(s_{i+1})}\prod_{i<n\text{ odd}}\frac{F_1(s_i)}{F_1(s_{i+1})} \\
&= F_1(s_n)\prod_{i=0}^{n-1}\frac{F_1(s_i)}{F_1(s_{i+1})} \qquad\qquad = F_1(s_0),
\end{aligned}
$$

which is independent of $x$.

*Part (b).* Because jointly optimal AFlowNet policies $P_1, P_2$ exist by Prop. 2, and they satisfy (10) by part (a) of this proposition, it suffices to show that the constraint (10) uniquely determines $P_1$ and $P_2$ for all pairs of reward functions $(R_1, R_2)$ for which $R_1(x)R_2(x) = \frac{1}{B_1(x)B_2(x)}$ for all $x \in \mathcal{X}$.

We prove this by strong induction on the number of states in $G$. The base case $|\mathcal{S}| = 1$ (there is a unique state which is both initial and terminal) is trivial: the products are empty and the constraint reads $Z = R_1(x)$.

Now suppose that the constraint uniquely determines $P_1$ and $P_2$ for all reward functions satisfying $R_1(x)R_2(x) = \frac{1}{B_1(x)B_2(x)}$ on graphs with fewer than $N$ states, for some $N > 1$, and consider an AFlowNet $G = (\mathcal{S}, \mathcal{A})$ with $N$ states, for which (10) holds and the reward functions satisfy $R_1(x)R_2(x) = \frac{1}{B_1(x)B_2(x)}$. It is easy to see that there exists a state $s \in \mathcal{S} \setminus \mathcal{X}$ such that all children of $s$ are terminal; select one such state $s$.

Suppose that $s \in \mathcal{S}_1$. We construct a new graph $G'$ and rewards $R_1', R_2'$ by deleting the children of $s$, making $s$ a terminal state, and modifying the reward function so that $R_1'(s) = \sum_{x\in\operatorname{Ch}(s)}R_1(s)$, setting $R_2'(s)$ so as to preserve $R_1'(s)R_2'(s) = \frac{1}{B_1(x)B_2(x)}$, and setting $R_i'(x) = R_i(x)$ for all other terminal states $x$. Thus the graph $G'$ is a two-player AFlowNet with alternating turns and satisfying the constraint on rewards.

We claim that the constraint (10) for $G, R_1, R_2$ and a pair of policies $P_1, P_2$ on $G$ implies the constraint for $G', R_1', R_2'$ and the same policies restricted to the states in $G'$. For all terminal states in $G'$ inherited from $G$, the constraint is unchanged. For the new terminal state $s$, we sum the constraints on $G$ for the children $x_1, \ldots, x_K \in \operatorname{Ch}(s)$. Letting $s_0 \to s_1 \to \cdots \to s_n = s$ be the trajectory leading to $s$, we have:

$$
\sum_{k=1}^{K}\left[Z\prod_{i<n:s_i\in\mathcal{S}_1}P_1(s_{i+1}\mid s_i)P_1(x_k\mid s)\right] = \sum_{k=1}^{K}\left[R_1(x_k)B_2(x_k)\prod_{i:s_i\in\mathcal{S}_2}P_2(s_{i+1}\mid s_i)\right]
$$

$$
\left(Z\prod_{i<n:s_i\in\mathcal{S}_1}P_1(s_{i+1}\mid s_i)\right)\sum_{k=1}^{K}P_1(x_k\mid s) = \left(B_2(s)\prod_{i:s_i\in\mathcal{S}_2}P_2(s_{i+1}\mid s_i)\right)\sum_{k=1}^{K}R_1(x_k)
$$

$$
Z\prod_{i<n:s_i\in\mathcal{S}_1}P_1(s_{i+1}\mid s_i) = B_2(s)\prod_{i:s_i\in\mathcal{S}_2}P_2(s_{i+1}\mid s_i)R_1'(s),
$$

which is precisely the constraint for $G'$ at the state $s$. So the constraint (10) is satisfied on $G'$.

Since $G'$ has fewer than $N$ states, by the induction hypothesis, $P_1$ and $P_2$ are uniquely determined on $G'$ and are therefore equal to the jointly optimal AFlowNet policies. It remains to show that $P_1(\cdot\mid s)$ is uniquely determined. Indeed, the only factor on the left side of (10) that varies between

children $x$ of $s$ is $P_1(x \mid s)$, while on the right side, the only such factor is $R_1(x)$. It follows that if the constraint is satisfied, then $P_1(x \mid s) \propto R_1(x)$, which uniquely determines $P_1(\cdot \mid s)$.

The case $s \in \mathcal{S}_2$ is analogous. $\qquad\square$

**Proposition 5.** *Suppose that $P_{\text{agent}}$ satisfies the EDB constraints. Then, for any $s \in \mathcal{S}_{\text{agent}}$ and $s' \in \text{Ch}(s)$,*

$$P_{\text{agent}}(s' \mid s) \propto \mathbb{E}_{G_{\text{env}} \sim P_{\text{env}}}\left[F^{G_{\text{env}}}(s') \mid s \in V(G_{\text{env}})\right],$$

*where the expectation is taken over the distribution over strategies determined by $P_{\text{env}}$, restricted to the strategies that contain the state $s$.*

*Proof of Proposition 5.* We first note that the expression inside the expectation is well-defined, since if $s \in V(G_{\text{env}}) \cap \mathcal{S}_{\text{agent}}$, then all children of $s$ are also in $V(G_{\text{env}})$.

Now suppose that $P_{\text{agent}}$ satisfies EDB jointly with a flow function $F$. By (4), we have $P_{\text{agent}}(s' \mid s) \propto F(s')$, and $s' \in V(G_{\text{env}})$ is equivalent to $s \in V(G_{\text{env}})$ for $s \in \mathcal{S}_{\text{agent}}$ and $s' \in \text{Ch}(s)$. Therefore, it would suffice to show that

$$F(s) = \mathbb{E}_{G_{\text{env}} \sim P_{\text{env}}}\left[F^{G_{\text{env}}}(s) \mid s \in V(G_{\text{env}})\right]$$

for all $s \in \mathcal{S} \setminus \mathcal{X}$.

To do so, we show that the expression on the right side satisfies the recurrence (13). We consider three cases:

- If $s \in \mathcal{S}_{\text{agent}}$, then the child set of $s$ in any $G_{\text{env}}$ containing $s$ is the same as its child set in $G$. It follows that

$$
\begin{aligned}
\mathbb{E}_{G_{\text{env}} \sim P_{\text{env}}}\left[F^{G_{\text{env}}}(s) \mid s \in V(G_{\text{env}})\right] &= \mathbb{E}_{G_{\text{env}} \sim P_{\text{env}}}\left[\sum_{s' \in \text{Ch}(s)} F^{G_{\text{env}}}(s') \mid s \in V(G_{\text{env}})\right] \\
&= \sum_{s' \in \text{Ch}(s)} \mathbb{E}_{G_{\text{env}} \sim P_{\text{env}}}\left[F^{G_{\text{env}}}(s') \mid s \in V(G_{\text{env}})\right] \\
&= \sum_{s' \in \text{Ch}(s)} \mathbb{E}_{G_{\text{env}} \sim P_{\text{env}}}\left[F^{G_{\text{env}}}(s') \mid s' \in V(G_{\text{env}})\right],
\end{aligned}
$$

 showing the first case of the recurrence.
- If $s \in \mathcal{S}_{\text{env}}$, then in any $G_{\text{env}}$ containing $s$, $s$ has a unique child $s'$ and $F^{G_{\text{env}}}(s) = F^{G_{\text{env}}}(s')$. We decompose the expectation into terms depending on the child of $s$ that is present in $G_{\text{env}}$:

$$
\begin{aligned}
\mathbb{E}_{G_{\text{env}} \sim P_{\text{env}}}\left[F^{G_{\text{env}}}(s) \mid s \in V(G_{\text{env}})\right] &= \mathbb{E}_{\substack{s' \in \text{Ch}(s) \\ s' \sim P_{\text{env}}(s' \in G_{\text{env}} \mid s \in G_{\text{env}})}} \mathbb{E}_{G_{\text{env}} \sim P_{\text{env}}}\left[F^{G_{\text{env}}}(s) \mid s' \in V(G_{\text{env}})\right] \\
&= \mathbb{E}_{s' \sim P_{\text{env}}(s' \mid s)} \mathbb{E}_{G_{\text{env}} \sim P_{\text{env}}}\left[F^{G_{\text{env}}}(s') \mid s' \in V(G_{\text{env}})\right],
\end{aligned}
$$

 which shows the second case of the recurrence.
- The case $s \in \mathcal{X}$ is simple, since $F(s') = F^{G_{\text{env}}}(s') = R(s')$ for all $G_{\text{env}}$ containing $s$.

$\qquad\square$

## C  GAME SPECIFICATION

### C.1  TIC-TAC-TOE

Two players alternate between placing X tiles and O tiles in a $3 \times 3$ grid. If any player reaches a board with three of their pieces connected by a straight line, they win. Although simplistic, tic-tac-toe has a small enough state space that the ground truth EDB-based flow values can be computed and compared to the learned values. This allows for the unique opportunity to verify that the AFlowNet has converged to the predicted stable point. The stable point can be found by recursively visiting each state in the game and backpropagating the rewards, flows, and probabilities.

## C.2    CONNECT-4

There are two players who alternate between placing yellow and red tokens (for the sake of simplicity, we use X tiles and O tiles) in a $6 \times 7$ grid (6 rows, 7 columns). Each token is placed at the top of the grid and falls to the lowest unoccupied point in the column. If any player reaches a board with four of their pieces connected in a straight line, they win. Connect-4 has a far larger state space than tic-tac-toe: it is quite difficult for even humans to learn, although the first player has been proven to have a winning strategy (Allis, 1988). As such, it is computationally infeasible to compute, or to store, the optimal policies at all states in the AFlowNet.

## D    TRAINING DETAILS

For 1, we collect trajectories as sequences of tuples (`state, mask, curr_player, action, done, log_reward`):

- `state`: the current state of the environment
- `mask`: binary mask of legal moves over the action space
- `curr_player`: the player whose turn it is to make the action
- `action`: the sampled action at the given state
- `done`: whether the action resulted in a terminal state
- `log_reward`: the log reward if `done`

As architecture, we use a convolutional neural network composed of residual blocks inspired by the AlphaZero architecture (Silver et al., 2018; Thakoor et al., 2016) with a few modifications. We remove the batch normalization layers as the population statistics varied too much between training and evaluation. We use only the policy head (using one for each side, *e.g.*, playing as "X" or "O") and increase the number of filters it has as recommended by Prasad et al. (2018). We replace ReLU activations by Leaky ReLU activations, reduce the number of residual blocks and reduce the number of filters in each block (128 instead of 256) as tic-tac-toe/Connect-4 are simpler than chess/Go. Finally, we include a single differentiable parameter log $Z$.

The main training hyperparameters are:

- `num_trajectories_epoch`: number of trajectories added to the buffer every epoch
- `batch_size`: batch size used for training and trajectory generation
- `num_steps`: number of optimization steps per epoch
- `replay_buffer_capacity`: the maximum capacity of the replay buffer
- `learning_rate`: learning rate for the policy network
- `learning_rate_Z`: learning rate for the log $Z$ (we find that a higher value helps training)
- `num_residual_blocks`: number of residual blocks in the architecture

Specific values are included in Table 2.

### D.1    TRAINING/EVALUATION POLICY

During training, to generate the training trajectories, we sample actions using the softmax of the policy logits of the AFlowNet with a temperature coefficient of 1.5. At test time, (*i.e.*, for the tournaments), when it is the AFlowNet's turn to play, we select the move corresponding to $\arg\max_{s' \in Ch(s)} P_i(s' \mid s)$, where $i$ is the index of the player to make a move at $s$.

### D.2    ALPHAZERO TRAINING

AlphaZero is trained as per the specifications of Thakoor et al. (2016). The batch size of AlphaZero is changed to 512 to match the batch size of AFlowNets. Training discrepancies include the number of examples gathered and retained over each iteration, which is significantly different between the AFlowNet implementation and AlphaZero. Naturally, it is quite difficult to compare AlphaZero and AFlowNets exactly because we do not have an MCTS analogue. Additionally, AlphaZero trains on single transitions whereas AFlowNets with TB loss train on entire trajectories. This leads to a discrepancy in what is considered a training example: a transition versus a trajectory. The number of Monte Carlo tree search iterations for AlphaZero is 25 for both tic-tac-toe and Connect-4.

| Hyperparameter | Tic-tac-toe | Connect-4 |
|---|---|---|
| num_trajectories_epoch | 10240 | 10240 |
| batch_size | 512 | 1024 |
| num_steps | 500 | 250 |
| replay_buffer_capacity | 10240 | 250000 |
| learning_rate | 1e-3 | 1e-3 |
| learning_rate_Z | 5e-2 | 5e-2 |
| num_residual_blocks | 10 | 15 |
| GPU | 1xRTX3090Ti | 1xRTX8000 |

Table 2: Hyperparameters used for training the AFlowNets on on tic-tac-toe and Connect-4

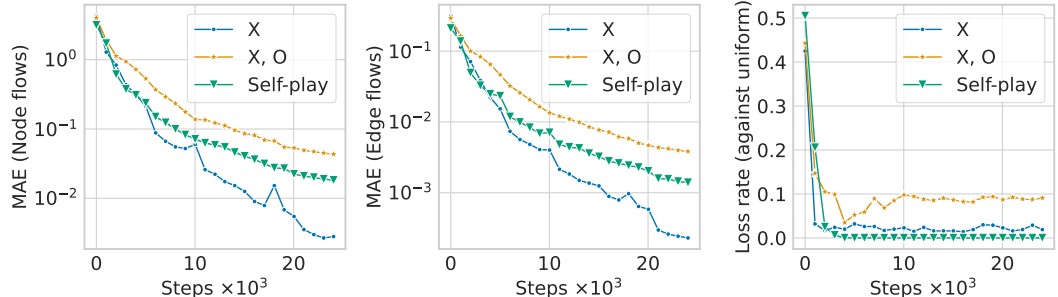

Figure 5: Graphs of learning performance over various training runs. (Left) The average MAE of learned node flows (not in log space) compared to ground truth flows computed algorithmically. (Middle) Average MAE for learned edge flows. (Right) Loss rate of the three training regimes against a random uniform opponent.

# E    ADDITIONAL RESULTS FROM ADVERSARIAL GAMES

## E.1    CONVERGENCE OF EFLOWNETS/AFLOWNETS TO UNIQUE OPTIMUM

In addition to testing game-playing performance, we aim to investigate whether EFlowNets/AFlowNets are capable of learning the correct flows (*i.e.*, those corresponding to the unique optimum). The game tree of tic-tac-toe is small enough that ground truth flows and policies (for a fixed opponent and the stable-point optimum) can be computed algorithmically by backtracking recursively from terminal states.

We train neural networks, see §D for details, to evaluate the following configurations:

(1) EFlowNet with a fixed stochastic opponent (policy consists of choosing each legal action with equal probability) from one perspective (*e.g.*, always plays X).
(2) EFlowNet versus a fixed stochastic opponent, learning both perspectives.
(3) AFlowNet learning using the EDB objective with off-policy self-play.

Figure 5 illustrates the learning performance of the three tested configurations. For all configurations, the ground truth flows/policies that satisfy the EDB constraints are learned, as evidenced by the left and middle graphs. Importantly, this is even the case for training through self-play, where the AFlowNet converges to the actual stable-point optimum.

Interestingly, the AFlowNet does not need to learn the exact flows to achieve strong game-playing performance. For example, here the AFlowNet is able to always win or draw against a uniform agent after less than 5000 steps even though its MAE relative to the correct flows still decreases substantially in the next 20000 steps. Additionally, the EFlowNet formulation is not enough to obtain a robust game playing agent, particularly when the agent it is playing against does not play well.

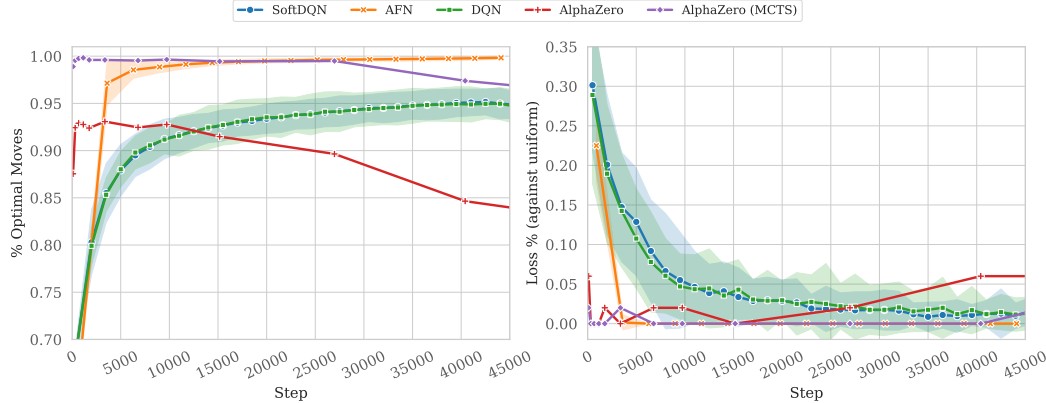

Figure 6: Graphs of learning performance over various training runs for AFN, DQN, SoftDQN and AlphaZero. **(Left)** The percent of optimal moves (for TicTacToe solved through minimax) over all states. **(Right)** Loss rate of the algorithms against a random uniform opponent.

### E.2    COMPARISON TO DQN AND SOFTDQN

We include additional results comparing DQN and SoftDQN to AFlowNets on TicTacToe in Fig. 6. Getting standard RL algorithms to work in multi-agent settings (learning through self-play) is non trivial and does not work in many common RL libraries.

To remedy this, and to ensure as fair of a comparison as possible, we implement DQN and its soft equivalent inside our framework (i.e. using the same environment as AFN, the same architecture, etc.). Notably, to achieve an agent that could consistently beat a uniform opponent, it was necessary to use a minimax version of DQN similar to (Fan et al., 2020) for sequential games (i.e. the q-update is based on the negation of the *maximum q-value of the opponent*). For the soft version (denoted SoftDQN), we sample trajectories using the softmax of the Q-values and similarly use the softmax for the updates (once again in a minimax fashion).

### E.3    TIC-TAC-TOE TOURNAMENT

To test the performance of AFlowNets against state of the art methods such as AlphaZero, we train a popular open-source AlphaZero implementation Thakoor et al. (2016) to play tic-tac-toe, pitting the agents against each other and baselines in a tournament. In the set of baselines we include a uniform opponent and a tree-search agent[2]. By changing the value of $\lambda$ for the AFlowNet, we also test how the learned policy changes with varying rewards. We proceed to test the performance of EDB-based AFlowNets and TB-based AFlowNets.

**Results with EDB-based AFlowNets**    Some selected results of the tournament are listed in Table 3. Note that AlphaZero is trained with Monte Carlo tree search (MCTS), but is tested in the tournament with MCTS both on and off. This ensures a fair comparison of inference-time game-playing capabilities as AFlowNets have no such tree-search mechanism to generate a policy.

It is clear from the tournament results in Table 3 that AlphaZero and AFlowNet agents are capable of perfect play in tic-tac-toe, drawing nearly every game that was played. The biggest differences in performance come from playing against the uniform and tree-search agents. It is clear that AFlowNet$_{10}$ performed the best, winning or drawing all games against the uniform and tree-search agents. Interestingly, while a higher $\lambda$ produces a better X-playing agent, the same is not true for agents playing second: against the tree-search and uniform agents, a lower $\lambda$ corresponds to a better score. As such, there may be a diminishing return with higher values of $\lambda$, perhaps encouraging overly-risky behaviour.

Training with MCTS seems to greatly improve the speed of convergence, as AlphaZero converges to a stable Elo after only a few training steps, see Figure 7. In comparison, AFlowNets require more training steps, optimization steps, and training examples to reach a similar level of performance

---

[2]The tree-search agent uses AlphaZero's value function, a search depth of three, and alpha-beta pruning.

Table 3: Selected results of AFlowNets (AFlowNet$_\lambda$) pitted against AlphaZero (A0) and baselines in tic-tac-toe. The agents listed in the rows are playing first as X, the agents listed in the columns are playing second as O. AFlowNets are trained five times with different seeds. The results of the tournament represent the mean and standard deviation of the games over the different seeds, where wins, draws, and losses are given two points, one point, and zero points respectively. Each element in the table is the result from the perspective of the X-playing agent in the row. For example, A0 playing X achieved a score of 44.2 ± 5.1 against the uniform agent.

| ×↓ \| ○→ | AFlowNet$_2$ | AFlowNet$_{10}$ | AFlowNet$_{15}$ | A0 | A0+MCTS | Uniform | Tree Search |
|---|---|---|---|---|---|---|---|
| AFlowNet$_2$ | – | 25 ± 0 | 35 ± 13.7 | 25 ± 0 | 25 ± 0 | 45.6 ± 1.5 | 45.4 ± 6.4 |
| AFlowNet$_{10}$ | 25 ± 0 | – | 35 ± 13.7 | 25 ± 0 | 25 ± 0 | 49.8 ± 1.1 | 47.6 ± 5.4 |
| AFlowNet$_{15}$ | 15 ± 13.7 | 15 ± 13.7 | – | 30 ± 20.9 | 23.2 ± 18.3 | 43 ± 2.6 | 36 ± 22.0 |
| A0 | 25 ± 0 | 25 ± 0 | 35 ± 13.7 | – | 25 ± 0 | 44.2 ± 5.1 | 50 ± 0 |
| A0+MCTS | 25 ± 0 | 25 ± 0 | 38.4 ± 13.7 | 25 ± 0 | – | 47.4 ± 2.1 | 49.8 ± 0.4 |
| Uniform | 7.2 ± 2.3 | 6.8 ± 0.5 | 13.8 ± 7.6 | 11 ± 7.5 | 10.6 ± 3.6 | – | 22.8 ± 5.6 |
| Tree Search | 0 ± 0 | 0 ± 0 | 15 ± 22.4 | 0 ± 0 | 0.4 ± 0.9 | 37.8 ± 3.6 | – |

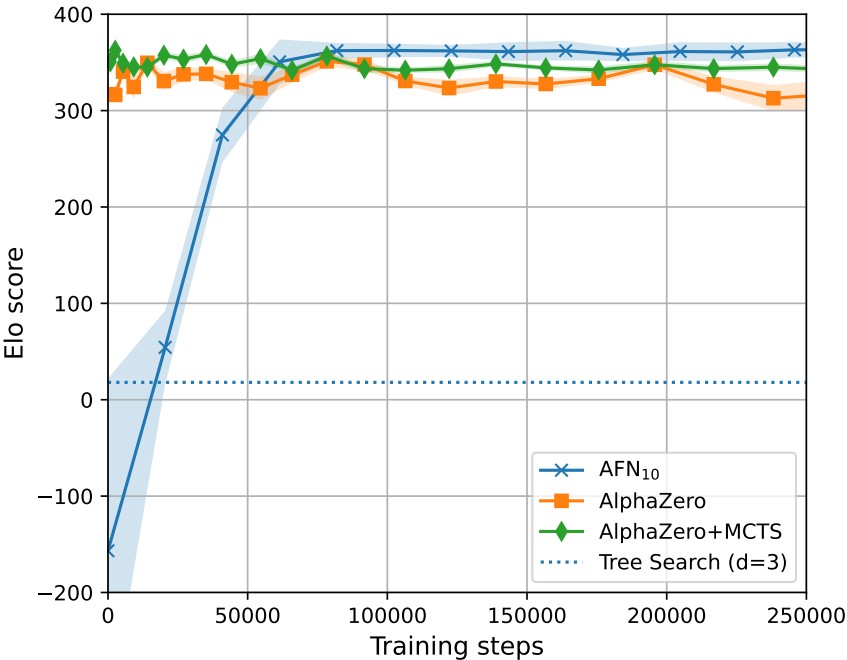

Figure 7: Elo over optimization steps while training a EDB-based AFlowNet agent in tic-tac-toe. The Elo of the agents after the full course of training are 363.2 ± 7.7, 357.2 ± 12.7, 356.9 ± 11.6, 344.3 ± 4.6, and 317.6 ± 14.3 for AFlowNet$_{10}$, AFlowNet$_2$, AFlowNet$_{15}$, AlphaZero+MCTS, and AlphaZero respectively. Other AFlowNet agents and error bars for the baseline models are omitted for clarity.

to AlphaZero in terms of Elo. AFlowNet$_{10}$ achieves the highest Elo, reinforcing its tournament performance in Table 3. Again, it appears that a larger $\lambda$ produces a worse AFlowNet agent, with AFlowNet$_{15}$ achieving the lowest Elo of all AFlowNets. The AlphaZero agents also achieve high Elo scores, with AlphaZero+MCTS achieving the better score of the two.

AlphaZero converges to a stable Elo after a small number of optimization steps whereas AFlowNets using the EDB constraint require an order of magnitude more steps to reach a similar Elo (about 50k steps versus 5k to 10k steps). A similar trend holds for training time, with AFlowNets requiring about 15 times longer to reach similar Elo scores (AlphaZero+MCTS took 38 seconds to reach an Elo of 46 whereas the first AFlowNet to reach a similar Elo of 51 took 558 seconds). While AFlowNets can certainly learn to play tic-tac-toe effectively, they clearly require far more training time and computation to achieve similar levels of performance to AlphaZero. We have not tested AlphaZero without MCTS in training, nor AFlowNets with tree search, so the comparison naturally favors AlphaZero given the power of tree search.

Table 4: Selected results of AFlowNets (AFlowNet$_\lambda$) pitted against AlphaZero (A0) and a random uniform baseline in Connect-4. The agents listed in the rows are playing first and the agents listed in the columns are playing second. AFlowNets are trained three times with different seeds. The results of the tournament represent the mean and standard deviation of the games over the different seeds, where wins, draws, and losses are given two points, one point, and zero points respectively. Each element in the table is the result from the perspective of the agent that played first in the row. For example, A0 playing first achieved a score of 49.2 ± 1.8 against the uniform agent.

| × ↓ │ ○ → | AFlowNet$_2$ | AFlowNet$_{10}$ | AFlowNet$_{15}$ | A0 | A0+MCTS | Uniform |
|---|---|---|---|---|---|---|
| AFlowNet$_2$ | – | 0 ± 0 | 5 ± 11.2 | 50 ± 0 | 32.8 ± 19.4 | 50 ± 0 |
| AFlowNet$_{10}$ | 50 ± 0 | – | 20 ± 27.4 | 50 ± 0 | 46.2 ± 8.5 | 50 ± 0 |
| AFlowNet$_{15}$ | 50 ± 0 | 35 ± 22.4 | – | 50 ± 0 | 50 ± 0 | 50 ± 0 |
| A0 | 0 ± 0 | 0 ± 0 | 0 ± 0 | – | 25 ± 0 | 49.2 ± 1.8 |
| A0+MCTS | 8.8 ± 8.7 | 0.8 ± 1.8 | 0 ± 0 | 25 ± 0 | – | 49 ± 1.0 |
| Uniform | 0 ± 0 | 0 ± 0 | 0 ± 0 | 8.4 ± 2.6 | 1.2 ± 1.6 | – |

**Results with TB-based AFlowNets**  The results thus far have focused on EDB-trained agents, but it is important to demonstrate the performance of TB-based agents as well. Figure 3 illustrates the Elo over three training runs with different seeds of AFlowNet$_{10}$ and the baseline models. Clearly, AFlowNet$_{10}$ matches the Elo of AlphaZero and converges quickly. Compared to Figure 7, it appears that TB-based agents converge about as quickly as AlphaZero, about 10 times faster than the EDB-based agents. Interestingly, the Elo of the TB-based AFlowNets is lower than the Elo of the EDB-based AFlowNets.

There appears to be little difference in the performance of a TB-trained AFlowNet with different values of $\lambda$ in tic-tac-toe. This is in contrast to the EDB-trained AFlowNets which seemed to be affected by the setting of $\lambda$. Similarly to the EDB-based agents, almost every game in the tournament was a draw, with small differences between the performance of an agent against the baselines dictating the differences in Elo.

### E.4    Connect-4 tournament

We also run a tournament in Connect-4 against AlphaZero and a uniform random baseline. Again, we vary $\lambda$ to test how the reward structure changes agent performance. The tournament results in Table 4 indicate that the effect of $\lambda$ is similar to the experiments in tic-tac-toe. A higher $\lambda$ produces a better agent, however the diminishing return in these experiments relates to a reduction in benefit when increasing lambda rather than a decrease in Elo. This supports the idea that the reward structure affects the behaviour of an agent. The results of the Connect-4 tournament corroborate the Elo results of Figure 3. The Elos of agents AFlowNet$_2$, AFlowNet$_{10}$, and AFlowNet$_{15}$ are 1190.8 ± 64.2, 1700.1 ± 60.0, and 1835.3 ± 154.9. Clearly, AFlowNet$_{15}$ is the best agent.

