# OpenReview forum: "Expected flow networks in stochastic environments and two-player zero-sum games"
_ICLR.cc/2024/Conference — ICLR 2024 poster_

### Official Review · Reviewer_C2iS · 2023-10-27

**Soundness:** 3 good
**Presentation:** 3 good
**Contribution:** 3 good
**Rating:** 6
**Confidence:** 3

**Summary:**

The paper proposes two extensions of generative flow networks (GFlowNets): expected flow networks (EFlowNets), designed to operate in stochastic environments, and adversarial flow networks (AFlowNets) for (two-player) zero-sum games. They experimentally demonstrate that EFlowNets outperform earlier approaches for protein design, and AFlowNets outperform AlphaZero in simple zero-sum games.

**Strengths:**

GFlowNets have shown promise in developing robust agents in challenging Markov decision processes. The paper provides a novel and natural formulation of GFlowNets to stochastic and adversarial environments. Unlike an earlier attempt by Pan et al., the new formulation satisfies a number of desirable theoretical properties, which is also reflected in the experimental results. For the adversarial setting I am not aware of any prior comparable formulations. Moreover, the experimental results also seem overall promising.

**Weaknesses:**

First, in terms of the experiments, the current results do not go far enough at least in the zero-sum setting. In particular, the games tested (tic-tac-toe and connect-4) are not large enough in order to make solid conclusions regarding the scalability of the method. It would be much more convincing if the authors used this new formulation to make progress on benchmarks that are otherwise elusive using prior techniques. Overall, I believe that the paper would significantly benefit from having more experiments and on larger benchmarks. It would also expect the adversarial formulation to capture partially observable settings (such as Poker); I do not see any significant obstacles in extending the methodology to partially observable settings. Such extensions would significantly strengthen the contributions of the paper. Besides those issues with the experiments, the other concern is that the new formulations are relatively straightforward, and there is not much conceptual or algorithmic novelty in deriving them based on earlier approaches. Overall, although the results are promising, they seem to be in a rather preliminary stage.

**Questions:**

What was the previous state of the art method for the protein design task considered in Section 4.1? Is it included in the current comparisons?

---

> ### Author Response · Authors · 2023-11-17
> **Response to Reviewer C2iS (1/2)**
>
> Thank you for your comments, concerns, and questions. We would like to first reiterate our contributions and the novelty of our constructions. As your review points out, our construction of flow networks in stochastic environments is novel and satisfies a number of previously neglected desiderata. The constraints we design for EFlowNets, the Expected Detailed Balance (EDB) constraints, depart from the usual constructions for GFlowNets. These novel constructions lead to an interesting and unique class of models: Expected Flow Networks (EFlowNets), which have not been previously proposed nor analyzed.
>
> The extension of EFlowNets to multi-agent environments yields a surprising finding: EFlowNets pitted against one another have a unique stable point optimum. This result is very exciting and a first of its kind: AFlowNets can be seen as inherently capable of self-play. Our further exploration of AFlowNets naturally leads from the EFlowNet criteria, leading to novel and non-trivial constraints (trajectory balance for AFlowNets, Proposition 4). These new TB constraints allow for efficient training and effective credit assignment in adversarial settings, leading us to propose another unique class of models: branch-adjusted Adversarial Flow Networks (AFlowNets).
>
> We thus strongly agree with your assessment of our constructions as novel and unique. The math being “natural” does not imply the results are straightforward to conjecture and prove, and we believe we have made important theoretical innovations.
>
> > First, in terms of the experiments, the current results do not go far enough at least in the zero-sum setting. In particular, the games tested (tic-tac-toe and connect-4) are not large enough in order to make solid conclusions regarding the scalability of the method. It would be much more convincing if the authors used this new formulation to make progress on benchmarks that are otherwise elusive using prior techniques. Overall, I believe that the paper would significantly benefit from having more experiments and on larger benchmarks.
>
> While we agree that running experiments on larger benchmarks would be interesting, we urge caution in this direction. While tic-tac-toe is a relatively simple game to learn, it still presents a real learning challenge to a model and can be used for illustrative purposes. Further, Connect-4 is non-trivial and requires substantial effort and computational resources to learn effectively. In our review of the literature we did not find any methods that learned to play Connect-4 in a multi-agent setting. We argue that this lack of prior work is an indication of the difficulty of learning such games (note also that it is quite difficult for a human to beat a trained AFlowNet agent at this game). As such, our method does make progress on benchmarks that were otherwise elusive using prior techniques (we **learn** to play the games, we do not use solvers or heuristics).
>
> Additionally, we have limited access to computational resources and cannot perform experiments that expend thousands of GPU hours. More complex games along the order of Chess and Go required thousands of TPUs [1] or GPUs [2] which is unfortunately far beyond reasonable computational capacities.
>
> The AlphaZero (AZ) algorithm is computationally intensive to train. This is predominantly due to the Monte Carlo Tree Search (MCTS) for evaluating a position when generating training examples. Each iteration of the AZ algorithm involves traversing the tree and performing inference using the neural network. Each evaluation of a position necessitates a non-trivial number of iterations. The number of iterations needed increases with the complexity of the environment [1]. In comparison, AFlowNets do not rely on MCTS and simply require a single pass of the neural network.
>
> > It would also expect the adversarial formulation to capture partially observable settings (such as Poker); I do not see any significant obstacles in extending the methodology to partially observable settings. Such extensions would significantly strengthen the contributions of the paper.
>
> We agree that an extension to partially observable settings would be an interesting research direction. We believe that the experiments we have shown in this work demonstrate a meaningful contribution to the community as fully observable games already attract a lot of interest. We will endeavor to tackle partially observable games in future work.

---

> > ### Comment · Reviewer_C2iS · 2023-11-19
> > **Thank you for your Response**
> >
> > I thank the authors for their response.
> >
> > In your response you are claiming that "While tic-tac-toe is a relatively simple game to learn, it still presents a real learning challenge to a model and can be used for illustrative purposes." In what sense is tic-tac-toe a real learning challenge? It can be trivially solved with min-max search.
> >
> > You are also claiming that "our method does make progress on benchmarks that were otherwise elusive using prior techniques." I don't really see this this. Connect-4 has been already solved with other methods. Overall you are making a distinction between "learning" and "solving" which is not entirely clear to me. Can you elaborate more on this point? The main advantage of your approach seems to be that you are reaching a high Elo in a less amount of time compared to other approaches, which is definitely a valuable contribution.
> >
> > I am also not entirely following your claim that "EFlowNets pitted against one another have a unique stable point optimum. This result is very exciting and a first of its kind." Isn't this in line with the minimax theorem that holds for any two-player zero-sum game? What is there specifically about AFlowNets that is special?

---

> ### Author Response · Authors · 2023-11-17
> **Response to Reviewer C2iS (2/2)**
>
> > Besides those issues with the experiments, the other concern is that the new formulations are relatively straightforward, and there is not much conceptual or algorithmic novelty in deriving them based on earlier approaches. Overall, although the results are promising, they seem to be in a rather preliminary stage.
>
> We again highlight our constructions as stated above. **The results are not obvious to conjecture and prove; for example, the existence of a TB loss for branch-adjusted AFlowNets was a big surprise to us** (note that it does not exist without the branching factor adjustment we propose, or if the unadjusted log-rewards do not sum to zero!). The beauty and simplicity of a piece of mathematics does not detract from its novelty, efficacy, and power.
>
> > What was the previous state of the art method for the protein design task considered in Section 4.1? Is it included in the current comparisons?
>
> The stochastic protein design task was introduced by [Pan et al., 2023]. The main previous methods for the task are included in the current comparisons. The red line represents Stochastic GFN, which introduced the task and which we argue is insufficient for effective sampling in stochastic environments. We give reasoning for the insufficiency of that method in Section 2.2, and our method outperforms theirs in peakier reward environments (higher beta), as evident in Figure 2.
>
> [1] D.Silver et al., "A general reinforcement learning algorithm that masters chess, shogi, and Go through self-play”, Science 362.6419 (2018): 1140-1144.
>
> [2] Y. Tian et al., "Elf opengo: An analysis and open reimplementation of alphazero”, ICML 2019.
>
> **We hope that this has addressed your questions. Please let us know if there is anything else we can clarify.**

---

> ### Author Response · Authors · 2023-11-19
> **Response 2 to Reviewer C2iS**
>
> We would like to thank the reviewer for their prompt response and their engagement in discussion. We are happy to answer the new questions below.
>
> > In your response you are claiming that "While tic-tac-toe is a relatively simple game to learn, it still presents a real learning challenge to a model and can be used for illustrative purposes." In what sense is tic-tac-toe a real learning challenge? It can be trivially solved with min-max search.
>
> We see the role of tic-tac-toe as an illustrative game where we can instantiate the full game tree and compute the optimal move at every state.
>
> Tic-tac-toe can be solved with exhaustive search, but the challenge is **amortizing** move selection with a learned policy trained through RL-style exploration (with stable training possible but also not trivial to achieve, for instance, with AlphaZero). Amortizing symbolic/programmatic computation and planning into a neural network is a common thread in ML research.
>
> > You are also claiming that "our method does make progress on benchmarks that were otherwise elusive using prior techniques." I don't really see this this. Connect-4 has been already solved with other methods. Overall you are making a distinction between "learning" and "solving" which is not entirely clear to me. Can you elaborate more on this point? The main advantage of your approach seems to be that you are reaching a high Elo in a less amount of time compared to other approaches, which is definitely a valuable contribution.
>
> Connect-4 (with 4 531 985 219 092 [1] possible legal states) is not practical to solve at runtime with min-max search; computation of optimal moves requires specially designed programs with human-constructed heuristics. Our method, despite only having been trained on an insignificant fraction of these states, learns to play effectively in the vast majority of them (Figure 4). Importantly, it does so without inference-time search and human heuristics. It relies solely on generating trajectories and observing the associated outcomes during training.
>
> **Our goal is learning (amortization), not solving (finding the exact optimal move).** As we show, state-of-the-art learning approaches to two-player games are less efficient and successful in this task than the one we propose. Notably, AlphaZero is far less competitive in Connect-4 unless the learned policy is **combined** with an approximate recursive search (MCTS), while AFlowNets simply sample the trained policy to get the next move. This underscores the difficulty of learning an amortized policy.
>
> We believe this ability to learn effective policies **without any form of inference-time search** is a particularly exciting aspect of our method.
>
> > I am also not entirely following your claim that "EFlowNets pitted against one another have a unique stable point optimum. This result is very exciting and a first of its kind." Isn't this in line with the minimax theorem that holds for any two-player zero-sum game? What is there specifically about AFlowNets that is special?
>
> There are two differences between our results and the minimax theorem (and between our algorithms and the minimax search for optimal strategy computation).
> - The minimax theorem is about perfect play (Nash equilibrium), while AFlowNets find a particular kind of quantal response equilibrium (see Appendix A). As we prove in Proposition 5, AFlowNet policies can also be expressed by a recursive computation, replacing max and min in min-max search by certain expectation operators.
> - Minimax is inherently a saddle point optimization problem and so requires an alternating optimization over the agents, such as via self-play, where each agent optimizes their policy independently against their opponent. On the other hand, AFlowNets use a single loss function (TB, Proposition 4)  to **jointly** optimize two stochastic policies, or mixed strategies. The exciting result is that this loss function has a unique optimum, and that this **joint** optimum recovers a QRE-optimal strategy for each of the players **independently**. This result is not obvious and rests on assumptions specific to two-player zero-sum games, and the ability to optimize the two policies jointly in this way is an advantage for credit assignment and training efficiency as demonstrated in our experiments.
>
> [1] “A212693.” *OEIS*, https://oeis.org/A212693. Accessed 19 Nov. 2023.

---

> > ### Author Response · Authors · 2023-11-20
> > **Follow-up**
> >
> > Dear Reviewer C2iS,
> >
> > Thank you again for your review and follow-up questions. We'd like to ask if you have any further questions before the end of the discussion period -- we would be happy to answer them! -- and whether our responses have affected your assessment of the paper.
> >
> > The authors.

---

> > ### Comment · Reviewer_C2iS · 2023-11-21
> > **Thank you for the Response**
> >
> > I thank the authors for their helpful response. I now understand better the contribution of the experiments in the adversarial setting, and the results are overall convincing. I have increased my score accordingly.

---

### Official Review · Reviewer_cwh7 · 2023-10-28

**Soundness:** 1 poor
**Presentation:** 2 fair
**Contribution:** 1 poor
**Rating:** 3
**Confidence:** 5

**Summary:**

This paper argues that it can address limitations of previous attempts for extending GFlowNets to stochastic transition dynamics by expected flow networks, which leads to meaningless solution. This paper also studies adversarial flow networks, to tackle the tree-structured MDP.

**Strengths:**

This paper studies an interesting setting about GFlowNets in tasks with stochastic transition dynamics.

**Weaknesses:**

- It is mentioned that "We propose expected flow networks, ... generalizing GFlowNets on tree-structured state spaces." In the simpler case of trees instead of non-tree DAGs, why don't you just apply maximum entropy reinforcement learning (MaxEnt RL) approaches? It has been extensively demonstrated in previous GFlowNets papers that when the DAG degenerates to a tree, it is indeed equivalent to MaxEnt RL approaches (e.g., Soft Q-learning), which can natually handle stochastic worlds. Therefore, it makes little sense for only EFlowNets to be only evaluated in tree-structured state spaces.

- The performance and explanation of EFlowNets does not make sense. As described in the paper, this paper aims to extend (Pan et al., 2023; Bengio et al., 2023) to handle more general cases considering stochasticity in the transition dynamics, i.e., extend GFlowNets -- which aims to sample proportionally to a reward function $R(x)$, i.e., $\pi(x) \propto R(x)$, to the settings with stochastic transition dynamics.
However, the solution learned by EFlowNets is incorrect as demonstrated in Figure 1(a). Although it is able to solve $p$ (with a solution of around $0.66$) in this case while $p$ cannot find satisfiable solutions in Stoch-GFlowNets, it leads to sampling $x1$ with probability $p(x_1) \approx 0.33 \neq 2/10$, $p(x_2) \approx 0.33 \neq 3/10$, $p(x_3) \approx 0.306 \neq 1/10$, and $p(x_2) \approx 0.034 \neq 4/10$. Although $p$ is solvable in EFlowNets, the learned sampling policy fails to realize the goal of GFlowNets (sampling proportionally from the reward function), and learns a meaningless sampling policy. Consequently, EFlowNets do not address the limitations of Stoch-GFlowNets in some cases where satisfiable solutions are unavailable. However, this claim is made throughout the paper, which is an overstatement in this context.

- The statement seems incorrect -- "Stochastic GFlowNets directly apply the training algorithms applicable to deterministic GFlowNets (e.g., DB) to the augmented DAG $G$, with the only modification being that the forward policy $P_F$ is free to be learned only on agent edges, while on environment edges it is fixed to the transition function." In stochastic environments, the transitions dynamic is unknown, and after I checked the Stochastic GFlowNets paper, I found that it indeed learns the forward policy and the underlying transition dynamics (which is not fixed, unknown and is learned by experiences).

- In the "Violated desiderata in stochastic GFlowNets" section, a very important property is missed (which is satisfied by Stochastic GFlowNets with solvable cases but EFlowNets unfortunately violated) -- the correct sampling behavior, which means that GFlowNets function well in stochastic environments which sample proportionally to the reward function. Failures to satisfy this basic and fundamental requirement will lead to meaningless solutions.

- EFlowNets bear great similarity to (Yang et al., 2023), however, the latter aims to extend Decision Transformer to stochastic environments in the realm of RL. Taking the expectation is correct in RL, since the objective of RL is to maximized the expected discounted future rewards. In addition, EFlowNets also bear similarity to Section 3.3.2 about stochastic rewards in (Bengio et al., 2023), which is mentioned about stochasticity in the rewars (not the transition dynamics), which is a case studied in (Zhang et al., 2023). Therefore, it does not make sense to apply the methodology in about learning an expectation as in RL (Yang et al., 2023) to GFlowNets with the hope to tackle stochasticity in transition dynamics as discussed in (Bengio et al., 2023).

- It is mentioned that trajectory balance is used for branch-adjusted AFlowNets. However, most practical games in multi-agent systems are actually partially observable (e.g., StartCraft II), which renders stochasticity in the transition dynamics. TB leads to large variance as mentioned in the text and in experiments in (Madan et al., 2023). Therefore, it seems to be more appropriate to apply this with DB or SubTB (since one of the claims for SubTB is to improve credit assignment).

- It is mentioned that "This data defines a fully observed sequential game", which can limit its practical applicability in a wider range of applications with partial observation.

- A very relevant paper extending GFlowNets to the multi-agent setting  (Li et al., 2023) is not cited and discussed. Since both paper study a very similar setting, it is worth comparing the approach with FCN and CFN in (Li et al., 2023), or at least discussed thoroughly. "Generative Multi-Flow Networks: Centralized, Independent and Conservation. Yinchuan Li, Haozhi Wang, Shuang Luo, Yunfeng Shao, Jianye Hao. 2023."

- It is unclear why experiments in Section 4.1 employ very large $\beta=10, 30$ which is different from (Jain et al., 2022), which leads to the case of a tree with very peaky rewards. In addition, it is worth investigating how the approach behave under the traditional $L_1$ error metric, which measures how well EFlowNets learn compared with other approaches.

- Why the two-play games correspond to a tree instead of a graph? There can be many parent states for a state actually.

**Questions:**

Please check the weakness part.

---

> ### Author Response · Authors · 2023-11-17
> **Response to Reviewer cwh7 (1/2)**
>
> We thank the reviewer for their time and in-depth review of our paper. We would like to address their points and clarify various aspects of our contribution that we believe have gone underappreciated.
>
> > In the simpler case of trees instead of non-tree DAGs, why don't you just apply maximum entropy reinforcement learning (MaxEnt RL) approaches? It has been extensively demonstrated in previous GFlowNets papers that when the DAG degenerates to a tree, it is indeed equivalent to MaxEnt RL approaches (e.g., Soft Q-learning), which can naturally handle stochastic worlds. Therefore, it makes little sense for only EFlowNets to be only evaluated in tree-structured state spaces.
>
> You are right that GFlowNets = MaxEnt RL in **deterministic** environments, but this is not the case in stochastic environments. To demonstrate this point, we train a soft version of DQN using the exact same architecture as AFlowNets on tic-tac-toe with self-play. After a non-negligible amount of hyperparameter tuning, we obtain an agent that learns a meaningful policy (winning >99% of the time against a uniform agent) but is still noticeably worse than AFlowNets and AlphaZero both in terms of training time and the policy it seems to converge to (worse % of optimal moves and still loses occasionally to a uniform agent). See the new results in Appendix E.2.
>
> > Although p is solvable in EFlowNets, the learned sampling policy fails to realize the goal of GFlowNets (sampling proportionally from the reward function), and learns a meaningless sampling policy. [...] Failures to satisfy this basic and fundamental requirement will lead to meaningless solutions.
>
> This is precisely the problem in stochastic environments: **learning a policy that samples proportionally to reward is in general impossible**. As illustrated in Fig. 1(a), even in an incredibly small tree, it is impossible to sample proportionally to reward (as you cannot change the environment dynamics). StochGFNs aim to sample from the reward distribution, but this is not always possible, as we show in that example. (This issue with StochGFNs has as a consequence that the global optimum may depend on the training policy or not be unique – it may be impossible to achieve zero loss.)
>
> **EFlowNets solve a fundamentally different problem** – sampling from the expectation over the environment’s transitions. This is one reason we invented a new name (**expected** flow networks), rather than calling them variants of GFlowNets, as generative flow networks should solve the generative modeling problem of sampling from the reward distribution.
>
> The advantage of the EFlowNet formulation is *satisfiability*: in the overwhelming majority of cases where sampling proportional to reward is not possible, the EFlowNet constraints are still satisfiable and the set of global optima of the loss does not depend on the choice of full-support training policy (as well as other benefits mentioned in the paper).
>
> In addition, we would like to politely push back against the assertion that the learned policy is meaningless. We believe considering your expected reward from a state is an intuitive way to deal with random transitions and show connections between EFlowNets and marginalized quantal response policies (see Appendix A).
>
> > Stochastic GFlowNets paper, I found that it indeed learns the forward policy and the underlying transition dynamics (which is not fixed, unknown and is learned by experiences).
>
> We agree this sentence is misleading and it will be amended. The point being made is that they use the deterministic GFlowNet algorithms with a learned policy for agent edges and a transition function for environment edges. This transition function can either be taken as given or learned through some maximum likelihood objective. Similarly, EFlowNets can either take the transition function as given or learn it (in the TFBind task, it is learned, just as in StochGFNs).
>
> >EFlowNets bear great similarity to (Yang et al., 2023) [...] Therefore, it does not make sense to apply the methodology in about learning an expectation as in RL (Yang et al., 2023) to GFlowNets with the hope to tackle stochasticity in transition dynamics as discussed in (Bengio et al., 2023).
>
> We are not sure about why the referenced works imply it is inappropriate to model uncertainty in transition dynamics in the way we do. Actually, [Bengio et al.] already mention that the modeling in [Pan et al., 2023] that aims to sample proportionally to reward is not in general satisfiable, which is one consideration that motivates us to consider a different sampling problem in stochastic environments – the one solved by EFlowNets.

---

> ### Author Response · Authors · 2023-11-17
> **Response to Reviewer cwh7 (2/2)**
>
> > Therefore, it seems to be more appropriate to apply this with DB or SubTB (since one of the claims for SubTB is to improve credit assignment).
>
> We thank the reviewer for pointing out the potential flaws of TB, but we found that the improved credit assignment of the TB objective already yielded strong results (in spite of the potential variance) when compared to the EDB constraints. (We also note that our TB for AFlowNets is distinct from the TB in GFlowNets.)
>
> This parallels the development of GFlowNets, where TB was first proposed as an improvement over DB for improved credit assignment [Malkin et al., 2022] and then SubTB was proposed for problems where TB has excessively high variance [Madan et al., 2023]. We agree that SubTB could potentially help further, especially in problems with longer trajectories, and believe this is an interesting future area of research!
>
> > It is mentioned that "This data defines a fully observed sequential game", which can limit its practical applicability in a wider range of applications with partial observation.
>
> We agree this is a limitation of the work, but the fully observable setting is widely studied in the RL / adversarial games literature. We already mention the extension to partially observable states as possible future work in the conclusion and continue to think this is an interesting future direction.
>
> > A very relevant paper extending GFlowNets to the multi-agent setting (Li et al., 2023) is not cited and discussed
>
> We thank the reviewer for pointing this paper out. The paper in question ([a rejected ICLR’23 submission](https://openreview.net/forum?id=OTIhUlChVaT)) discusses a setting that is not applicable to turn-taking zero-sum games, as it considers multiple agents taking actions simultaneously. We can mention this paper in the related work, but its relevance to our problems is quite limited and experimental comparison is not appropriate.
>
> > It is unclear why experiments in Section 4.1 employ very large Beta.
>
> In the StochGFN paper, the beta parameter was artificially set to induce a less flat sampling distribution. We experiment with larger betas **in addition to** to the baseline value as this makes the problem of sampling from the reward distribution unsatisfiable (since addition of randomness induces smoothness in the sampling distribution, while the reward is peaky).
>
> > Why the two-play games correspond to a tree instead of a graph? There can be many parent states for a state actually.
>
> Indeed most two-player games are graphs instead of trees; however, any DAG can be converted to a tree-structured environment by augmenting states with history. See the end of Section 2.1 for further details.
>
> **We hope that we have addressed your concerns. Please let us know if you have any further questions.**

---

> > ### Author Response · Authors · 2023-11-20
> > **Follow-up**
> >
> > Dear Reviewer cwh7,
> >
> > Thank you again for your review. We'd like to ask if you have any further questions before the end of the discussion period -- we would be happy to answer them! -- and whether our responses have affected your assessment of the paper.
> >
> > The authors.

---

### Official Review · Reviewer_g2xq · 2023-11-01

**Soundness:** 3 good
**Presentation:** 3 good
**Contribution:** 3 good
**Rating:** 8
**Confidence:** 3

**Summary:**

This paper proposes expected flow networks, which extend generative flow networks to stochastic environments. They also extend their formulation to adversarial environments for two-player zero-sum games. Experiments show that the proposed methods achieve promising empirical performance.

**Strengths:**

This paper proposes a new formulation for learning in stochastic environments, which ensures that a set of desiderata can always be satisfied. The propose modification is intuitive and elegant. It can also be generalized to zero-sum game in a straightforward way, making it a widely applicable formulations. The writing is mostly clear and it gives a good introduction.

**Weaknesses:**

The paper still requires solid background knowledge on generative flow networks and it is not very friendly to non-expert.

**Questions:**

1. From Figure 3, it seems that the proposed method is much more computation heavy, can you show the computation time requires for each step of each method? Also how does the computation time scale with the size of the problem for each method?

---

> ### Author Response · Authors · 2023-11-17
> **Response to Reviewer g2xq (1/2)**
>
> We would like to thank you for your thoughtful review and insightful questions. Our responses to your questions and concerns are below.
>
> > From Figure 3, it seems that the proposed method is much more computation heavy, can you show the computation time requires for each step of each method? Also how does the computation time scale with the size of the problem for each method?
>
> It is important to compare the adversarial algorithms under a common framework. When developing the training scripts, we took inspiration from AlphaZero-like methods [1]. In this training framework, an iteration is defined by the following steps:
> 1. Collect training data by interacting with the environment. AlphaZero employs MCTS to construct training examples; we can instead just sample whole trajectories so long as the sampling policy has full support.
> 2. Train the neural networks on the training data for a number of epochs. (Comparing epoch counts is somewhat misleading as AlphaZero trains policy and value networks on single transitions, while we train a policy network and scalar $\log Z$ on whole trajectories.)
> 3. Evaluate performance. Pit agents against baselines or old agents [1].
>
> AlphaZero and AFlowNets are trained for 100 iterations each. The data points in Figure 3 represent the performance measured every five or so iterations. An important note on the seeming performance differential: AlphaZero collects a small number of examples on each iteration and adds them to a buffer [1, 2]. Our implementation did not do this, instead we sample about 10 times more examples on every iteration and fully replace our old examples. Also note that our implementation collects whole trajectories, not individual transitions. As such, the data collection process is quite computationally expensive, but can be made more efficient in future implementations by implementing a replay buffer similar to AlphaZero.
>
> The presence of a buffer alone, however, does not fully capture the performance differential. An important factor to consider is the use of MCTS in AlphaZero, a recursive algorithm which runs 25 or more game simulations for each step in a trajectory [2]. The more complex the game, the more simulations need to be run to maintain good performance.
>
> Assuming that the number of trajectories collected for our method and the number of transitions collected for AlphaZero (once the buffer is full) are equal, the training process takes roughly the same amount of time to complete.
>
> We can roughly state the worst-case computational complexity of AlphaZero data collection as $O(NMT)$. We can roughly state the computational complexity of AFlowNet data collection as $O(NT)$. The variables are:
> - $N$: number of examples to collect,
> - $M$: number of MCTS simulations,
> - $T$: maximum length of a trajectory.
>
> The computational complexity of collecting data for AlphaZero is expected to grow substantially with more complex games as $M$ must be increased. We believe that the computational complexity of training given a similar number of training examples is similar for both methods. The computational complexity of evaluation should also be relatively similar. As such, we expect the computational complexity of data collection to dominate. We list data collection times below for various settings of $M$ and $N$ in tic-tac-toe. $T = 9$ in the worst case.
> - AlphaZero: 17 seconds to collect examples, 4 seconds to train for 10 epochs. $M=25$, $N = 5224$.
> - AlphaZero: 24 seconds to collect examples, 4 seconds to train for 10 epochs. $M=50$, $N = 5736$.
> - AlphaZero: 42 seconds to collect examples, 4 seconds to train for 10 epochs. $M=100$, $N = 6216$.
> - AlphaZero: 72 seconds to collect examples, 4 seconds to train for 10 epochs. $M=200$, $N = 5864$.
> - AFlowNet: 8 seconds to collect examples, 4 seconds to train for 10 epochs. $N = 6144$.
>
> It is clear from the above that, for the same number of examples, the AFlowNet data generation process is quicker than AlphaZero’s data generation process. Furthermore, the training time is roughly similar between the two methods. We did not directly compare evaluation time because the evaluation process is quite different between the two methods, and serves more as a benchmark of progress rather than a necessary part of training. Thus, as the game gets more complex, we expect AFlowNets to have a faster data generating process than AlphaZero, allowing our method to train on more data in the same amount of time.

---

> ### Author Response · Authors · 2023-11-17
> **Response to Reviewer g2xq (2/2)**
>
> Another concern is sample efficiency. We believe that MCTS-based examples are more informative than raw trajectories. This may be why AlphaZero is able to converge much more quickly than AFlowNets in tic-tac-toe and Connect-4. We expect, however, that the disparities in convergence speed will be made up for in the number of examples we are able to collect and train on, perhaps leading to similar, if not better, convergence speed for more complex games.
>
> We note that it is quite difficult to compare the methods exactly, but we trust that the above explanation should suffice as motivation for AFlowNets and their potential efficiency.
>
> > Paper is not friendly to non-experts.
>
> We concede that the paper is fairly dense, but believe we have given the minimal background on GFlowNets that is required for understanding our generalizations. We would be glad to consider any recommendations which would make the paper more beginner-friendly for those less familiar with GFlowNets!
>
> [1] D.Silver et al., "A general reinforcement learning algorithm that masters chess, shogi, and Go through self-play”, Science 362.6419 (2018): 1140-1144.
> [2] S.Thakoor et al., “Learning to play othello without human knowledge”, https://github.com/suragnair/alpha-zero-general.
>
> **Thank you again for your review. Please let us know if you have more questions, and we will be glad to answer them.**

---

> > ### Author Response · Authors · 2023-11-20
> > **Follow-up**
> >
> > Dear Reviewer g2xq,
> >
> > Thank you again for your review. We'd like to ask if you have any further questions before the end of the discussion period. We would be happy to answer them.
> >
> > The authors.

---

### Official Review · Reviewer_QVYG · 2023-11-04

**Soundness:** 2 fair
**Presentation:** 2 fair
**Contribution:** 2 fair
**Rating:** 5
**Confidence:** 4

**Summary:**

The paper proposes Expected Flow Networks (EFlowNets) extending GFlowNets to stochastic environments. Stepping forward, Adversarial Flow Networks (AFlowNets) is introduced for adversarial environments like two-player zero-sum games. The authors show that EFlowNets outperform other GFlowNet formulations in tasks such as protein design and AFlowNets outperform AlphaZero in Connect-4.

**Strengths:**

1. The paper introduces EFlowNets and AFlowNets as extensions of GFlowNets, which provide solutions for generative modeling in stochastic and adversarial environments respectively.
2. The paper provides theoretical analysis and derives training objectives for EFlowNets and AFlowNets. The experiments conducted demonstrate the effectiveness of the proposed methods. The demo is also interesting.

**Weaknesses:**

1. The differences between stochastic environments and common environments for GFlowNet may not be stated clearly.
2. The line of extension is similar to that in reinforcement learning. However, not enough comparisions to RL algorithms are provided in the experiments in both environments.

**Questions:**

1. Can you provide more clear description of the differences between stochastic environments and common environments?
2. What is the motivation of extending GFlowNet to stochastic environments and adversarial environments? And what is the general advantage of GFlowNet-based methods compared to RL methods in both environments?
3. How do you think about the potential of AFlowNet compared to RL methods considering the limitation in the proposed algorithms currently?

---

> ### Author Response · Authors · 2023-11-17
> **Response to Reviewer QVYG (1/2)**
>
> We would like to thank the reviewer for their insightful comments on the paper. We would like to quickly articulate the novelty and power of our constructions for applications in stochastic and adversarial settings. Existing GFlowNet literature is relatively sparse when it comes to environment stochasticity. Those methods that do exist emphatically stick to the deterministic GFlowNet construction despite this construction providing undesirable sampling behavior or unsatisfiable constraints, which are not appropriate to our settings (see Figure 1(a)).
>
> In this paper, we depart from the deterministic constructions of GFlowNets in favor of a new paradigm of flow networks: Expected Flow Networks (EFlowNets) and Adversarial Flow Networks (AFlowNets). Our new constraints, expected detailed balance (EDB) and especially the branch-adjusted trajectory balance (TB), are an algorithmic novelty. They have nontrivial proofs of correctness, are grounded in the notions of quantal response policies (Appendix A), and allow for the first time the inference of stochastic equilibria in zero-sum games without self-play.
>
> We respond to your questions and concerns in the following paragraphs.
>
> > Can you provide more clear description of the differences between stochastic environments and common environments?
>
> In common (deterministic) environments, a transition from one state to another is wholly determined by the action taken at the starting state. For example, if we have a state $s_0$ with two corresponding actions: $a_1$ and $a_2$; which transition to states $s_1$ and $s_2$ respectively, by committing action $a_2$, the agent is certain to transition to $s_2$. Under our notation, this corresponds to the environmental transition $P_{\text{env}}(s_2|(s_0,a_2)) = 1$. In other words: when we take an action, we know where we will end up.
>
> In contrast, stochastic environments feature stochasticity in their transitions. If we extend the above example so that $P_{\text{env}}(s_2|s_0) \leq 1$, the action $a_2$ may not lead to $s_2$ every time, depending on the environment’s transition function. In AFlowNets, the environment’s transition function is seen as an adversary’s policy, $P_{\text{adv}}(-)$, **allowing us to model adversarial games as a special type of stochastic environment**.
>
> The distinction between deterministic and stochastic environments is important for GFlowNets in particular since the classical GFlowNet theory assumes that transitions are deterministic (constructive actions applied to an object give known results).
>
> > What is the motivation of extending GFlowNet to stochastic environments and adversarial environments? And what is the general advantage of GFlowNet-based methods compared to RL methods in both environments?
>
> Since GFlowNets are theorized as operating in deterministic environments, it is natural to extend them to stochastic environments. Some interesting examples of real-world stochasticity are 1) demand-response for electricity grids where the variation of the energy supply comes from the weather [1], 2) market making for securities [2], and 3) supply chain management [3].
>
> The main motivation for extending GFlowNets specifically to stochastic and adversarial environments is their ability to sample a diverse set of high-reward objects. In stochastic environments, our goal is to maintain a diverse sampling of high-reward outcomes despite the inherent unpredictability. In the case of adversarial environments, we aim to produce agents which can play with a variety of winning strategies, rather than overfitting to a single mode that produces high rewards but is also exploitable. In both cases, it is natural to sample states according to their expected reward rather than the raw reward as there is no longer any guarantee that this reward will be reached given a set of actions taken in an environment.
>
> Both of these aims are in stark contrast to the usual paradigm of reinforcement learning, which is to maximize rewards. Existing literature has demonstrated that GFlowNets are capable of sampling a more diverse set of solutions than comparable RL methods [4, 5].

---

> ### Author Response · Authors · 2023-11-17
> **Response to Reviewer QVYG (2/2)**
>
> > How do you think about the potential of AFlowNet compared to RL methods considering the limitation in the proposed algorithms currently?
>
> We believe that our method has considerable advantages in zero-sum games such as tic-tac-toe and Connect-4, especially compared to standard RL methods (e.g., PPO, Deep Q-Learning). The latter struggle to learn strong policies through self-play in these multi-agent settings. To demonstrate this, after significant tuning, we manage to train a DQN and SoftDQN agent (through self-play) on tic-tac-toe that can fairly consistently beat the uniform agent. Even so, they take considerably longer to train and converge to a suboptimal solution. **We have added results with DQN and SoftDQN agents to Appendix E.2.**
>
> [1] V.Mai et al, “Multi-Agent Reinforcement Learning for Fast-Timescale Demand Response of Residential Loads”, AAMAS 2023.
>
> [2] T.Spooner et al., "Robust market making via adversarial reinforcement learning”, ICJAI 2020.
>
> [3] F.Stranieri and F.Stella, "A deep reinforcement learning approach to supply chain inventory management", arXiv:2204.09603.
>
> [4] E.Bengio et al. "Flow network based generative models for non-iterative diverse candidate generation”, NeurIPS 2021.
>
> [5] M.Jain et al. "Biological sequence design with GFlowNets”, ICML 2022.
>
> **We hope to have addressed all of your concerns. We are happy to answer any further questions you may have.**

---

> > ### Author Response · Authors · 2023-11-20
> > **Follow-up**
> >
> > Dear Reviewer QVYG,
> >
> > Thank you again for your review. We'd like to ask if you have any further questions before the end of the discussion period -- we would be happy to answer them! -- and whether our responses have affected your assessment of the paper.
> >
> > The authors.

---

### Author Response · Authors · 2023-11-17
**General response and new RL results**

We thank all reviewers for their comments. We respond to each of them below, but first we want to emphasize some aspects of the novelty in our work that some reviewers seem to have missed.
- **The formulation of EFlowNets:** We depart from deterministic GFlowNets (which aim to sample terminal states proportionally to the reward) and past stochastic GFlowNets (which aim to sample states proportionally to reward. Crucially, we would like to reiterate that it is in general **impossible to sample proportionally to reward in stochastic environments**. This leads to generally unsatisfiable constraints for stochastic GFNs, see Fig. 1(a). Instead, we propose a formulation grounded in the theory of quantal response agents, which features satisfiable objectives and is appropriate for our applications of interest.
- **Trajectory balance in AFlowNets:** This loss and the nontrivial result establishing its correctness (Proposition 4) is a major algorithmic novelty. It enables full-trajectory credit assignment and thus **finds stochastic equilibria in two-player zero-sum games without self-play**.
- **Comparisons to standard RL methods:** We would like to emphasize the challenge of learning strong policies even in seemingly simple games such as tic-tac-toe and Connect-4 given a reasonable compute budget. Standard RL algorithms (e.g., PPO, Deep Q-Learning) struggle to learn strong policies through self-play in these multi-agent settings. To demonstrate this, after significant tuning, we manage to train a DQN and SoftDQN agent (through self-play) on tic-tac-toe that can fairly consistently beat the uniform agent. Even so, they take considerably longer to train and converge to a suboptimal solution. **We have added results with DQN and SoftDQN agents to Appendix E.2.**
- **Difficulty of games:** While AlphaZero has been successful at learning strong policies from scratch for complex zero-sum sequential board games such as chess, to do so it requires enormous computational resources (on the order of thousands of TPUs/GPUs for chess [1]). Thus, learning effective policies from scratch for these games with a reasonable computational budget is still an open problem. Our results on Connect-4, showing AFlowNets performing strongly compared to AlphaZero with a reasonable computation budget, are a promising step in the direction of developing agents for larger games with more accessible resources.

[1] D.Silver et al., "A general reinforcement learning algorithm that masters chess, shogi, and Go through self-play”, Science 362.6419 (2018): 1140-1144.

---

> ### Author Response · Authors · 2023-11-22
> **RL results on Connect 4**
>
> We include additional results on Connect4 below. We attempt to train a DQN, A2C and PPO agent using an open-source repository [1] based on StableBaselines3 [2]. For PPO, we test 2 configurations: self-play and fixed training against a strong heuristic agent. After 72 hours of training on an A100, we take the best model and pit it against our trained AFN agent. Below are the results for 1000 games (A2C not included as it was unable to learn a meaningful policy):
>
> | Adversary       | First Player    | Result (for AFN): W/L/D |
> | --------------- | --------------- | ----------------------- |
> | PPO (Heuristic) | AFN             | 1000 / 0 / 0            |
> |                 | PPO (Heuristic)  | 995 / 0 / 5             |
> | PPO (Self play) | AFN             | 1000 / 0 / 0            |
> |                 | PPO (Self play) | 994 / 0 / 6             |
> | DQN (Self play) | AFN             | 1000 / 0 / 0            |
> |                 | DQN (Self play) | 997 / 0 / 3             |
>
> In addition, we compare with a specially-trained open-source version of AlphaGo Zero (with a deterministic policy) trained by the repository’s owner on Connect-4. The resulting game is a draw.
>
> Finally, we include the aforementioned tic-tac-toe results (i.e. % of optimal moves in tic-tac-toe) in table form for easier viewing:
>
> |Method \ Training Steps| **5000** | **10000** | **50000** |
> |------|------|------|------|
> | **DQN** | 88.02 ± 0.94 | 91.17 ± 0.76 | 92.21 ± 1.05 |
> | **SoftDQN** | 85.82 ± 0.99 | 89.99 ± 0.90 | 92.13 ± 1.40 |
> | **AlphaZero** | 92.46 ± NA | 91.48 ± NA | 82.68 ± NA |
> | **AlphaZero + MCTS** | **99.54 ± NA** | **99.45 ± NA** | 96.04 ± NA |
> | **AFN** | 97.14 ± 1.15 | **98.86 ± 0.31** | **99.83 ± 0.08** |
>
> [1] Raffin, Antonin, et al. "Stable baselines3." (2019).
>
> [2] "Connect Four Environment”. *Github*, https://github.com/lucasBertola/Connect-4-Gym-env-Reinforcement-learning.

---

### Meta-Review · Area_Chair_uAWr · 2023-12-06

**Metareview:**

I thank the authors for the robust discussion, their engagement, and prompt revision of the preprint. While reviewer cwh7 was the most critical, they were unfortunately unable to engage with the authors during the rebuttal period to confirm that the authors clarified some important points successfully. I think that the authors did a good job at responding to the various points, and my opinion of the paper is overall positive.

My biggest reservation for this paper is the comparison with soft Q-Learning, which in tree-form settings should be fundamentally equivalent to the approach, and indeed even from the new plots the authors included in the revised version performs basically the same as their approach. However, it seems that the authors' approach has other advantages, so overall I'm recommending acceptance.

**Justification For Why Not Higher Score:**

My biggest reservation for this paper is the comparison with soft Q-Learning, which in tree-form settings should be fundamentally equivalent to the approach.

**Justification For Why Not Lower Score:**

I believe that the paper has some merits, and the approach could see real adoption.

---

### Decision · Program_Chairs · 2024-01-16

Accept (poster)